# Klein Hyperbolic Metric Learning

Yulin Zhang [1]   Boxuan Hu [1]   Huimin Li [1]   Junlin Hu [1]

## Abstract

Hyperbolic metric learning is highly effective in embedding hierarchical data structures. However, past work has predominantly focused on the conformal Poincaré model, leaving other geometries like the Klein model largely under-explored. In addition, the curved geodesics of the Poincaré model present a fundamental geometric misalignment with the linear projections dominating the feature transformation steps in the modern neural network backbones. In this paper, we investigate the Klein model, a projective model of hyperbolic geometry whose straight-line geodesics offer a structurally aligned alternative in modern encoders, for hyperbolic metric learning. By formalizing a framework based on Einstein gyrovector operations, we derive a numerically stable metric learning approach that mitigates the inherent optimization challenges of the Klein model. Extensive experiments on multiple image datasets for fine-grained image classification task show that the Klein model not only serves as a viable alternative to the Poincaré model but also achieves highly competitive performance by leveraging its unique geometric properties, without increasing parameter complexity. Our empirical findings establish the Klein model as an efficient geometric prior for hyperbolic metric learning.

## 1. Introduction

Deep metric learning (DML) aims to learn a mapping function that projects raw data into an embedding space where distances accurately reflect semantic similarity. With the rise of deep learning, neural network-based DML has been widely adopted in image retrieval (Passalis et al., 2020; Suresh et al., 2021), biometric recognition (Hu et al., 2014; Ahmad & Fuller, 2019; Shao & Zhong, 2022), and anomaly detection (Ramachandra et al., 2021; Ju et al., 2020). Encoder architectures have evolved from CNNs to vision Transformers (ViTs), demonstrating superior scalability on large-scale datasets (Dosovitskiy et al., 2021). However, despite these architectural advancements, the learned representations remain largely confined to traditional Euclidean spaces, prompting frontier research to shift focus toward the exploration of the embedding manifolds themselves (Yan et al., 2021; Ermolov et al., 2022; Hong et al., 2024; Xu et al., 2026).

The geometric properties of the embedding space dictate the capacity to model complex hierarchical data. Euclidean spaces struggle to represent the tree-like hierarchies prevalent in nature, whereas hyperbolic spaces can embed such structures with minimal distortion due to their exponential volume growth (Sarkar, 2011). However, current hyperbolic metric learning methods are almost entirely dominated by the conformal Poincaré model (Ermolov et al., 2022; Yue et al., 2024; Li et al., 2025). Although the Poincaré model preserves angles, its geodesics are curved arcs. This curvature presents a significant geometric misalignment with the linear projection operations inherent in modern neural backbones, potentially leading to representational distortion.

In contrast, the Klein model, a projective model of hyperbolic geometry, features geodesics that are straight lines in the Euclidean sense. This pivotal property allows Euclidean-based vision encoders to map linear feature representations directly onto hyperbolic geodesics, significantly reducing distortion during the projection from Euclidean to hyperbolic space. Through this structural alignment, the Klein model provides a hyperbolic vehicle that is more consistent with the inductive bias of standard Euclidean backbones. Figure 1 shows the geometric differences between the Klein and Poincaré models.

In this paper, we systematically explore the application of the Klein model in hyperbolic metric learning. Our primary contributions are as follows:

- We introduce the Klein model of hyperbolic geometry to hyperbolic metric learning for fine-grained image classification, significantly expanding the application boundaries of deep metric learning in hyperbolic space.

- We propose a computational framework based on Ein-

---

[1]School of Software, Beihang University, Beijing, China. Correspondence to: Junlin Hu <hujunlin@buaa.edu.cn>.

*Proceedings of the 43rd International Conference on Machine Learning*, Seoul, South Korea. PMLR 306, 2026. Copyright 2026 by the author(s).

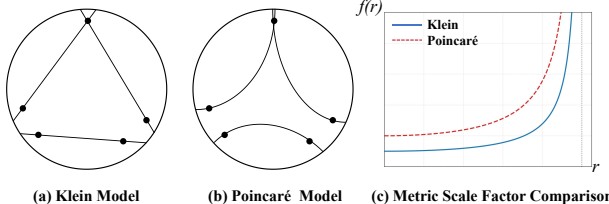

**(a) Klein Model**  **(b) Poincaré Model**  **(c) Metric Scale Factor Comparison**

*Figure 1.* Comparison of geometric and metric properties between Klein and Poincaré models. (a) Klein model shows that the hyperbolic geodesics (lines) are represented by straight Euclidean chords. (b) Poincaré model shows that the hyperbolic geodesics are represented by circular arcs orthogonal to the model boundary. (c) Metric scale factor comparison displays the radial metric scale factor $f(r)$ over radial distance $r$ for both models. The Klein model exhibits a less aggressive metric expansion.

stein gyrovector space. By formalizing the algebraic operations, we overcome the inherent numerical instability of the Klein model within computational graph.

• We conduct extensive experiments across four image benchmarks for fine-grained image classification task, validating the superiority of the Klein model as a geometric prior for hierarchical metric learning without increasing parameter complexity.

## 2. Related Work

### 2.1. Deep Metric Learning

The core objective of deep metric learning (DML) is to leverage deep neural networks to learn a feature embedding function that maps data into a metric space, where samples from the same class are pulled together while those from different classes are pushed apart (Ghojogh et al., 2022). The success of this objective is deeply coupled with the encoder architecture and the design of the loss function. Traditionally, encoders predominantly relied on classic CNN-based architectures such as ResNet (He et al., 2016), Inception (Szegedy et al., 2015), and DenseNet (Huang et al., 2017) to extract features with strong local perception. However, the advent of vision Transformers (ViTs) has revolutionized DML by leveraging self-attention mechanisms to aggregate global context (Wolf et al., 2020). From a geometric perspective, ViTs operate through a series of linear projections and global interactions; while providing superior scalability, this operational nature imposes specific structural requirements on the geometry of the embedding manifold.

The learning process is guided by loss functions, which are broadly categorized into pair-based and proxy-based families (Mohan et al., 2023). Pair-based losses, such as the classic contrastive and triplet losses, operate directly on sample embeddings. To enhance the discriminative power of

these methods, various improvements have been proposed. One line of work introduces geometric margins to enforce stricter separability constraints within the embedding space, such as angular or additive margins (Deng et al., 2019; Wang et al., 2018; Liu et al., 2017). Concurrently, to address the redundancy of easy samples, sophisticated sampling strategies like hard negative mining (Schroff et al., 2015) and distance-weighted sampling (Wu et al., 2017) have been developed to focus optimization on informative pairs. These methods excel at capturing fine-grained semantic relationships but are often constrained by a computational complexity of $O(N^2)$ or higher, which hampers training efficiency. In contrast, proxy-based losses introduce learnable "proxies" as class representatives, transforming the metric learning problem into a scalable classification task. While this significantly reduces computational overhead, it may come at the cost of overlooking subtle, fine-grained interactions between individual samples.

### 2.2. Hyperbolic Deep Learning

Hyperbolic deep learning leverages the exponential volume expansion and other non-Euclidean geometric properties of hyperbolic space to effectively model data with inherent hierarchical or tree-like structures. While traditional Euclidean methods often struggle with hierarchical distortion, hyperbolic embeddings have demonstrated superior capacity in representing structural priors, notably in modeling complex taxonomies. This geometric paradigm has proven globally effective, powering advancements in hyperbolic graph neural networks (Liu et al., 2019), hyperbolic variational autoencoders (HVAEs) (Mathieu et al., 2019), and natural language processing tasks (Valentino et al., 2024), where capturing structural depth is paramount.

As the cornerstone of this field, hyperbolic neural networks (HNN) generalize standard operations to their hyperbolic counterparts through gyrovector spaces (Ganea et al., 2018). This formalism provides algebraic foundation for different hyperbolic models, such as Möbius calculus for Poincaré model and Einstein calculus for Klein model. Recent advances have applied these geometric priors to various tasks, including few-shot learning (Khrulkov et al., 2020) and uncertainty modeling (Mandica, 2025). However, existing methods mainly rely on the conformal Poincaré model whose curved geodesics present a geometric misalignment with the linear operations inherent in modern neural architectures. In contrast, the Klein model, as a projective model characterized by straight-line geodesics, provides a promising yet under-explored alignment scheme for deep metric learning. While recent work has explored learning curvature (Fan et al., 2025) and feature augmentation (Gao et al., 2022) to enhance hyperbolic embeddings, it largely operates within Poincaré framework. Our work is orthogonal to these advances, focusing instead on the projective geometry itself.

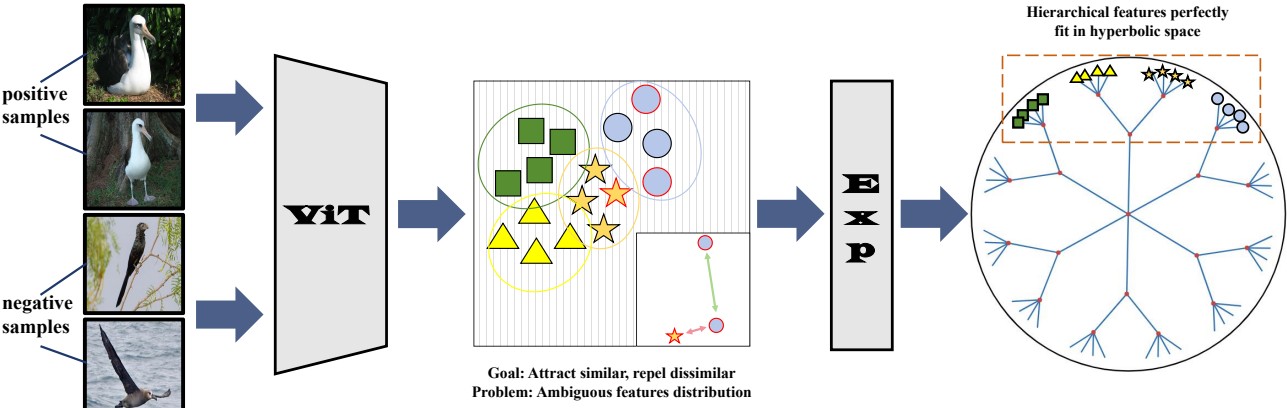

*Figure 2.* Flowchart of hyperbolic metric learning utilizing the Klein model. A vision Transformer (ViT) backbone first extracts Euclidean features from input images. The central example illustrates the challenges of metric learning in Euclidean space, which may lead to a cluttered feature distribution. This issue is highlighted in the bottom-right corner, where a dissimilar/negative pair is incorrectly positioned in close proximity, while a similar/positive pair remains far apart, directly contradicting the learning objective. Euclidean features are then projected into the Klein model using an exponential map (EXP). The resulting hyperbolic embedding effectively resolves these ambiguities by capturing the hierarchical relationships, leading to a well-structured representation with clear inter-class separation.

## 3. Methodology

This section details our Klein model-based hyperbolic metric learning method. Figure 2 shows a schematic overview.

### 3.1. Preliminary

Formally, an $n$-dimensional hyperbolic space $\mathbb{H}^n$ equipped with a hyperbolic metric is a Riemannian manifold with constant negative curvature $-c$. While prior research has predominantly focused on the conformal Poincaré model $(\mathbb{P}^n_c, ds^2_{\mathbb{P}})$, we adopt the Klein model $(\mathbb{K}^n_c, ds^2_{\mathbb{K}})$ (Cannon et al., 1997) as our embedding space due to its unique projective properties. The Klein model is represented as an $n$-dimensional ball $\mathbb{K}^n_c = \{\mathbf{x} \in \mathbb{R}^n : c\|\mathbf{x}\|^2 < 1, c > 0\}$, equipped with the Riemannian line element: $ds^2_{\mathbb{K}} = \frac{\|\mathbf{dx}\|^2}{1-c\|\mathbf{x}\|^2} + \frac{c\langle\mathbf{x},\mathbf{dx}\rangle^2}{(1-c\|\mathbf{x}\|^2)^2}$ (Cannon et al., 1997). Compared to the Poincaré metric, whose Riemannian line element is given by $ds^2_{\mathbb{P}} = \frac{4\|\mathbf{dx}\|^2}{(1-c\|\mathbf{x}\|^2)^2}$, the Klein metric exhibits a more gradual spatial expansion rate. This theoretical characteristic ensures a smoother transition from the center to the boundary, fostering a more stable environment for gradient-based optimization in deep models.

A pivotal geometric feature of the Klein model is that its geodesics coincide with Euclidean straight-line segments. To formalize this projective geometry, we utilize the framework of Einstein gyrovector spaces (Ungar, 2008). The algebraic structure is governed by Einstein addition as

$$\mathbf{x} \oplus_E \mathbf{y} = \frac{1}{1 + c\mathbf{x}^\mathsf{T}\mathbf{y}}\left(\mathbf{x} + \frac{1}{\gamma_\mathbf{x}}\mathbf{y} + c\frac{\gamma_\mathbf{x}}{1 + \gamma_\mathbf{x}}(\mathbf{x}^\mathsf{T}\mathbf{y})\mathbf{x}\right),$$

(1)

where $\gamma_\mathbf{x} = \frac{1}{\sqrt{1-c\|\mathbf{x}\|^2}}$ represents the Lorentz factor. To-

gether with Einstein scalar multiplication:

$$r \otimes_E \mathbf{x} = \frac{1}{\sqrt{c}}\tanh(r\operatorname{artanh}(\sqrt{c}\|\mathbf{x}\|))\frac{\mathbf{x}}{\|\mathbf{x}\|},$$

(2)

$(\mathbb{K}^n_c, \oplus_E, \otimes_E)$ constitutes an Einstein gyrogroup. This formalism captures the non-associative nature of relativistic velocity addition and provides a rigorous algebraic basis for our metric learning framework.

The standard distance between two points $\mathbf{x}, \mathbf{y}$ in the Klein model is given by

$$D^{\mathbb{K}}(\mathbf{x}, \mathbf{y}) = \frac{1}{\sqrt{c}}\operatorname{arcosh}\left(\frac{1 - c\mathbf{x}^\mathsf{T}\mathbf{y}}{\sqrt{1 - c\|\mathbf{x}\|^2}\sqrt{1 - c\|\mathbf{y}\|^2}}\right).$$

(3)

While mathematically sound, Equation (3) is prone to numerical instability in computational graphs due to the $\operatorname{arcosh}$ operator. To mitigate this, we employ the equivalent gyrovector distance:

$$D^{\mathbb{K}}_{\operatorname{gyr}}(\mathbf{x}, \mathbf{y}) = \frac{1}{\sqrt{c}}\operatorname{artanh}\left(\sqrt{c}\|-\mathbf{x} \oplus_E \mathbf{y}\|\right).$$

(4)

By leveraging the structure of Einstein addition, Equation (4) avoids the vanishing gradient issues associated with Equation (3), ensuring superior numerical stability during the training of deep metric models. The equivalence between Equation (3) and Equation (4) is given in Appendix A.1. A simple case study demonstrating the superior numerical stability of Equation (4) over Equation (3) during training is provided in Appendix A.2.

Notably, Equation (4) maintains the correspondence principle; as the curvature $c \to 0$, the Einstein gyrovector distance smoothly recovers the Euclidean metric, $D^{\mathbb{K}}_{\operatorname{gyr}}(\mathbf{x}, \mathbf{y}) \to$

$\|\mathbf{x} - \mathbf{y}\|$. This ensures that our framework serves as a geometric generalization that remains consistent with traditional linear representations in the limit of zero curvature.

To bridge the Euclidean vision encoder and the hyperbolic manifold, we employ the exponential map $\exp_{\mathbf{0}} : T_{\mathbf{0}}\mathbb{K}_c^n \to \mathbb{K}_c^n$ to project linear feature vectors into the Klein model (Mao et al., 2024) as:

$$\exp_{\mathbf{0}}(\mathbf{v}) = \tanh\left(\sqrt{c}\|\mathbf{v}\|\right)\frac{\mathbf{v}}{\sqrt{c}\|\mathbf{v}\|}. \tag{5}$$

### 3.2. Pairwise Cross-Entropy Loss

In each training iteration, we construct batches using an $N$-way, $m$-shot sampling strategy. Specifically, we sample $m$ instances from each of $N$ distinct classes, resulting in a total batch size of $K = mN$.

To optimize the embedding space, we minimize the PCE loss using the Einstein gyrovector distance $D_{gyr}^{\mathbb{K}}$. Unlike the Poincaré model where distance calculation involves highly non-linear conformal factors, the Klein model aligns distances along straight Euclidean lines. This geometric consistency fosters a more direct optimization landscape for the pairwise objectives defined on the manifold. For a positive pair of embedded points $\mathbf{z}_i$ and $\mathbf{z}_j$, the loss is defined by

$$\mathcal{L}_{i,j} = -\log\frac{\exp\left(-D_{\text{gyr}}^{\mathbb{K}}(\mathbf{z}_i, \mathbf{z}_j)/\tau\right)}{\sum_{k=1, k \neq i}^{K}\exp\left(-D_{\text{gyr}}^{\mathbb{K}}(\mathbf{z}_i, \mathbf{z}_k)/\tau\right)}, \tag{6}$$

where $\tau$ is the temperature hyperparameter. Then, the total loss $\mathcal{L}_{\text{PCE}}$ is calculated by summing the losses over all positive pairs $(i, j)$ as

$$\mathcal{L}_{\text{PCE}} = \sum_{(i,j)\in\mathcal{P}}\mathcal{L}_{i,j}. \tag{7}$$

### 3.3. Hierarchical Regularization Loss

We also exploit and extend the hierarchical regularization (HIER) loss proposed in (Kim et al., 2023). The primary objective of this formulation is to investigate the capability of the Klein model in constructing a mixed-curvature metric space with hyperspherical space. This setup allows us to comprehensively evaluate how effectively the Klein model can encode complex hierarchical prior knowledge. Furthermore, it serves as a meaningful test to verify the plug-and-play viability of the Klein geometry within joint metric learning frameworks.

We construct a composite loss function that integrates both hyperspherical space and the Klein model of hyperbolic space. In the hyperspherical domain, a conventional proxy-anchor metric learning loss $\mathcal{L}_{\text{ML}}$ (Kim et al., 2020) is employed. Its purpose is to leverage angular information to enhance inter-class separability. In the Klein model, we

introduce learnable proxies $\mathcal{Q}_{\text{HIER}} = \{\boldsymbol{\rho}_1, \boldsymbol{\rho}_2, ..., \boldsymbol{\rho}_n\}$ to capture latent semantic hierarchies. During the training process, triplets $\{\mathbf{z}_i, \mathbf{z}_j, \mathbf{z}_k\}$ are constructed using a reciprocal nearest neighbor mechanism. For each triplet, the model samples two lowest common ancestor (LCA) proxies from the set $\mathcal{Q}_{\text{HIER}}$ at different hierarchical levels: a related LCA $\boldsymbol{\rho}_{ij}$ representing the relationship between the related pair, and a triplet LCA $\boldsymbol{\rho}_{ijk}$ signifying a higher-level ancestral relationship for the entire triplet. Unlike the curved paths in the Poincaré model, the straight-line geodesics in the Klein model provide a structurally aligned trajectory for hierarchical anchoring. This ensures that the hierarchical prior effectively guides the embedding alignment without the geometric distortion inherent in curved-geodesic manifolds.

The core of the hierarchical loss $\mathcal{L}_{\text{HIER}}$ is formulated as

$$\begin{aligned}\mathcal{L}_{\text{HIER}} = \ &\max(D_{\text{gyr}}^{\mathbb{K}}(\mathbf{z}_i, \boldsymbol{\rho}_{ij}) - D_{\text{gyr}}^{\mathbb{K}}(\mathbf{z}_i, \boldsymbol{\rho}_{ijk}) + \delta, 0) \\ &+ \max(D_{\text{gyr}}^{\mathbb{K}}(\mathbf{z}_j, \boldsymbol{\rho}_{ij}) - D_{\text{gyr}}^{\mathbb{K}}(\mathbf{z}_j, \boldsymbol{\rho}_{ijk}) + \delta, 0) \\ &+ \max(D_{\text{gyr}}^{\mathbb{K}}(\mathbf{z}_k, \boldsymbol{\rho}_{ijk}) - D_{\text{gyr}}^{\mathbb{K}}(\mathbf{z}_k, \boldsymbol{\rho}_{ij}) + \delta, 0),\end{aligned} \tag{8}$$

where $\delta$ is a margin. The total loss function is a linear combination of conventional metric learning loss $\mathcal{L}_{\text{ML}}$ and HIER loss $\mathcal{L}_{\text{HIER}}$ as

$$\mathcal{L}_{\text{total}} = \mathcal{L}_{\text{ML}} + \lambda\mathcal{L}_{\text{HIER}}, \tag{9}$$

where the hyperparameter $\lambda$ serves as a weighting factor to balance the contributions of these two terms.

### 3.4. Stabilizing Hyperbolic Embeddings

As pointed out in (Guo et al., 2022), hyperbolic neural networks may suffer from vanishing gradients when embeddings are pushed towards the boundary of the Poincaré model. This issue is particularly relevant for the commonly used hybrid architectures that combine a Euclidean encoder with a hyperbolic projection head. The Klein model is also susceptible to this problem. This vulnerability arises because the gradient for Euclidean parameters is obtained by scaling the Riemannian gradient with the inverse of the Riemannian metric tensor. For the Klein model, the scaling factor is given by

$$\mathbf{G}^{\mathbb{K}}(\mathbf{z})^{-1} = (1 - c\|\mathbf{z}\|^2)(\mathbf{I}_n - c\mathbf{z}\mathbf{z}^{\mathsf{T}}). \tag{10}$$

The detailed derivation of this scaling factor utilizing the Sherman-Morrison formula is provided in Appendix A.3. As embeddings approach the boundary, this factor diminishes to zero, leading to the vanishing gradient problem. To mitigate this problem, we adopt a feature clipping technique (Guo et al., 2022) used for the Poincaré model and apply it to our Klein model. The clipping operation is defined as

$$\mathbf{x}_{\text{C}}^{\text{E}} = \min\left\{1, \frac{r}{\|\mathbf{x}^{\text{E}}\|}\right\}\mathbf{x}^{\text{E}}, \tag{11}$$

where $\mathbf{x}^{\text{E}}$ is the feature vector projected from the Euclidean encoder, $\mathbf{x}_C^{\text{E}}$ is its clipped counterpart, and $r$ is the clipping radius within the Klein model.

While hyperbolic models generally suffer from vanishing gradients at the boundary, the Klein model exhibits a more moderate metric expansion. As illustrated in Figure 1(c), the Klein scaling factor increases significantly more gradually than its Poincaré counterpart. Consequently, to capture a comparable effective embedding volume, the optimal feature clipping radius $r$ in Equation (11) naturally shifts to a larger value, utilizing a more extensive region of the Euclidean embedding ball.

## 4. Experiments

We follow the widely established training and evaluation scheme from (Kim et al., 2020). To evaluate our method, we conduct comprehensive experiments on four benchmark datasets designed for fine-grained image classification task.

### 4.1. Datasets

**CUB-200-2011** (CUB) (Welinder et al., 2010) is a fine-grained dataset introduced by Caltech and serves as a fundamental benchmark for fine-grained visual categorization. It contains 11,788 images across 200 different species of birds. Each image is annotated with class labels, bounding boxes, part locations, and binary attributes. For our experiments, the dataset is split by class, with the first 100 classes (5,864 images) used for training and the remaining 100 classes (5,924 images) reserved for testing. **Stanford Cars** (Cars) (Krause et al., 2013), also known as Cars-196, was released by the Stanford AI Lab in 2013 for fine-grained classification tasks. The dataset comprises 16,185 images of 196 classes of automobiles, where classes are defined by make, model, and year. Following the standard protocol, the first 98 classes (8,054 images) constitute the training set, while the other 98 classes (8,131 images) form the test set. **Stanford Online Products** (SOP) (Song et al., 2016) is a large-scale dataset widely used in metric learning and image classification research. It consists of 120,053 images of 22,634 online products sourced from eBay.com. The standard split uses the first 11,318 classes (59,551 images) for the training set and the remaining 11,316 classes (60,502 images) for the test set. **In-shop Clothes** (In-Shop) (Liu et al., 2016) is a subset of the DeepFashion dataset and is commonly used for in-shop clothes classification. It features multiple images for each clothing item, captured from various angles and viewpoints. The dataset contains 52,712 images of 7,982 products, with each image annotated with 463 attributes. The training set is composed of the first 3,997 classes (25,882 images). The test set contains the remaining 3,985 classes (26,830 images), which are typically partitioned into query and gallery sets for evaluation.

### 4.2. Backbone Architectures

We employ the vision Transformer (ViT) (Dosovitskiy et al., 2021) as the backbone architecture for feature extraction, and three ViT architecture-based pre-trained models are utilized.

**ViT-S** (Steiner et al., 2022) is used as the primary encoder in our experiments. This model consists of a 12-layer Transformer architecture with a hidden dimension of 384, 6 attention heads, and approximately 22 million parameters. While maintaining a relatively compact model size, ViT-S efficiently processes image data. The specific checkpoint used in our experiments is pre-trained on ImageNet-21k.

**DeiT-S** (Touvron et al., 2021) is a variant of ViT with a comparable size of approximately 22 million parameters and an identical architecture to ViT-S. DeiT introduces a knowledge distillation strategy that incorporates an additional distillation token. This token interacts with the class token and patch tokens through the self-attention layers and is trained to reproduce the hard-label predictions of a high-performing teacher network. The DeiT-S model is pre-trained on ImageNet-1k.

**DINO** (Caron et al., 2021) is a self-supervised learning framework that also leverages a student-teacher distillation approach, but without requiring any labels. DINO effectively prevents model collapse through a combination of multi-crop augmentation, centering, and sharpening of the teacher network's outputs. This allows it to learn powerful visual representations from unlabeled data. The DINO model is pre-trained on ImageNet-1k dataset without using any labels.

Following the feature extraction by the ViT-based backbone, we map the resultant vectors $\mathbf{v} \in \mathbb{R}^d$ into the Klein manifold $\mathbb{K}_c^n$ via the exponential map $\exp_{\mathbf{0}}$ defined in Equation (5).

A significant advantage of this architecture is the linear-projective alignment: since the final layer of a standard Transformer is a linear projection and the geodesics of the Klein model are Euclidean straight lines, the backbone's output space is naturally congruent with the manifold's shortest paths. This alignment eliminates the geometric distortion typically encountered in conformal model, where the curved geodesics create a structural mismatch with the linear operations of modern neural networks. We provide further empirical verification in Appendix B.2 to compare the structural distortion between these two hyperbolic models and the linear projection layers of modern backbones.

### 4.3. Implementation Details

All experiments are implemented using the PyTorch framework and conducted on a server with four NVIDIA GeForce RTX 4090 GPUs. We utilize ViT-S as the backbone encoder,

*Table 1.* Performance comparison (%) of the Klein model using pairwise cross-entropy (PCE) loss. Performance is evaluated using the Recall@$K$ metric with values indicated below. This table compares the Klein, Poincaré (Poinc) (Ermolov et al., 2022), and hyperspherical (Sph) (Ermolov et al., 2022) models for metric learning. We assess performance at two stages to analyze both the geometric embedding and the upstream encoder features. For the 128-dimensional embedding space, evaluation on test data relies on the intrinsic distance function of each geometry. For the 384-dimensional encoder features, we evaluate their performance on test data by computing cosine similarity on their $l_2$-normalized representations. Klein-DeiT-S, Klein-DINO, and Klein-ViT-S denote our methods using different ViT architectures. Methods marked with an asterisk (*) denote results reproduced by us using the official open-source code.

| Method | Dim | CUB | | | | Cars | | | | SOP | | | | In-shop | | | |
|---|---|---|---|---|---|---|---|---|---|---|---|---|---|---|---|---|---|
| | | 1 | 2 | 4 | 8 | 1 | 2 | 4 | 8 | 1 | 10 | 100 | 1000 | 1 | 10 | 20 | 30 |
| Sph-DeiT-S* | 128 | 72.4 | 81.6 | 88.0 | 93.0 | 73.6 | 82.9 | 88.6 | 92.8 | 76.2 | 89.3 | 95.9 | 98.8 | 88.4 | 96.7 | 97.6 | 98.1 |
| Poinc-DeiT-S* | 128 | 74.6 | 84.3 | 90.2 | 94.0 | 79.1 | 87.3 | 92.3 | 95.7 | 82.4 | 93.1 | 97.3 | 99.2 | 90.8 | 98.0 | 98.6 | 98.9 |
| **Klein-DeiT-S** | 128 | 73.2 | 82.9 | 89.3 | 93.5 | 82.3 | 89.8 | 94.1 | 96.6 | 83.4 | 93.5 | 97.5 | 99.2 | 92.2 | 98.2 | 98.8 | 99.0 |
| Sph-DINO* | 128 | 76.1 | 84.3 | 90.0 | 94.1 | 78.5 | 85.8 | 90.8 | 94.3 | 78.0 | 90.0 | 96.1 | 98.9 | 88.8 | 96.7 | 97.6 | 98.0 |
| Poinc-DINO* | 128 | 77.8 | 86.2 | 91.4 | 94.8 | 83.5 | 89.9 | 93.8 | 96.4 | 84.0 | 93.8 | 97.5 | 99.3 | 92.2 | 98.2 | 98.8 | 99.0 |
| **Klein-DINO** | 128 | 77.0 | 84.9 | 90.4 | 94.5 | **86.9** | **92.7** | **95.9** | **97.3** | 84.4 | 93.9 | 97.5 | 99.2 | 93.1 | 98.5 | **99.0** | 99.1 |
| Sph-ViT-S* | 128 | 83.3 | 89.6 | 93.8 | 95.9 | 74.0 | 83.5 | 89.2 | 93.3 | 79.6 | 91.4 | 96.8 | 99.2 | 89.6 | 97.3 | 98.1 | 98.4 |
| Poinc-ViT-S* | 128 | 83.0 | **90.3** | **94.1** | **96.2** | 80.5 | 88.2 | 93.1 | 95.9 | 84.9 | 94.5 | 98.0 | **99.4** | 92.5 | 98.4 | 98.9 | **99.2** |
| **Klein-ViT-S** | 128 | **83.4** | 89.9 | 93.8 | 96.1 | 83.4 | 90.4 | 94.5 | **97.3** | 85.8 | 95.0 | 98.2 | 99.4 | 93.5 | 98.6 | 99.0 | 99.2 |
| Sph-DeiT-S* | 384 | 75.2 | 84.1 | 89.8 | 93.7 | 78.9 | 87.0 | 92.1 | 95.3 | 77.4 | 89.7 | 95.8 | 98.8 | 88.8 | 96.7 | 97.7 | 98.1 |
| Poinc-DeiT-S* | 384 | 78.1 | 86.3 | 91.8 | 94.8 | 84.2 | 91.1 | 95.1 | 97.2 | 83.1 | 93.2 | 97.3 | 99.2 | 90.7 | 97.8 | 98.4 | 98.7 |
| **Klein-DeiT-S** | 384 | 76.4 | 85.1 | 90.5 | 94.4 | 85.7 | 92.0 | 95.5 | 97.7 | 83.4 | 93.3 | 97.4 | 99.2 | 91.4 | 97.8 | 98.5 | 98.8 |
| Sph-DINO* | 384 | 79.1 | 87.0 | 91.8 | 94.9 | 83.4 | 89.7 | 93.7 | 96.2 | 79.4 | 90.8 | 96.2 | 98.9 | 89.6 | 96.9 | 97.7 | 98.1 |
| Poinc-DINO* | 384 | 81.0 | 88.3 | 92.8 | 95.5 | 87.5 | 92.9 | 96.0 | 97.8 | 84.6 | 94.1 | 97.7 | 99.3 | 91.8 | 98.1 | 98.7 | 99.0 |
| **Klein-DINO** | 384 | 79.6 | 87.3 | 91.9 | 95.2 | **89.2** | **94.3** | **96.7** | **98.2** | 84.9 | 94.2 | 97.6 | 99.2 | **93.0** | 98.4 | **99.0** | 99.1 |
| Sph-ViT-S* | 384 | 83.7 | 90.4 | 94.1 | 96.3 | 78.9 | 86.7 | 91.8 | 95.2 | 79.8 | 91.3 | 96.6 | 99.0 | 89.8 | 97.2 | 98.0 | 98.3 |
| Poinc-ViT-S* | 384 | 85.0 | **91.2** | 94.4 | **96.5** | 84.9 | 91.0 | 94.9 | 97.3 | 85.2 | 94.5 | 98.0 | 99.4 | 92.4 | **98.4** | 98.9 | 99.1 |
| **Klein-ViT-S** | 384 | **85.1** | 91.1 | **94.5** | **96.5** | 86.5 | 92.4 | 95.8 | 98.0 | **85.7** | 94.7 | **98.1** | **99.5** | 93.0 | 98.4 | 98.9 | **99.1** |

*Table 2.* Performance comparison (%) of the Klein model using the hierarchical regularization (HIER) loss. Performance is evaluated using the Recall@$K$ metric with values indicated below. This table compares the effectiveness of using the Klein model versus the original Poincaré model (Kim et al., 2023) as the hierarchical regularization term. Both are applied to a base hyperspherical model trained with proxy-anchor loss. The dimensions 128 and 384 refer to the dimensionality of the final fused feature embedding. HIER-Klein-DeiT-S, HIER-Klein-DINO, and HIER-Klein-ViT-S denote our methods using different ViT architectures.

| Method | Dim | CUB | | | | Cars | | | | SOP | | | | In-shop | | | |
|---|---|---|---|---|---|---|---|---|---|---|---|---|---|---|---|---|---|
| | | 1 | 2 | 4 | 8 | 1 | 2 | 4 | 8 | 1 | 10 | 100 | 1000 | 1 | 10 | 20 | 30 |
| HIER-Poinc-DeiT-S | 128 | 75.2 | 84.2 | 90.0 | - | 85.1 | 91.1 | 95.1 | - | 82.5 | 92.7 | 97.0 | - | 91.0 | 98.0 | 98.6 | - |
| **HIER-Klein-DeiT-S** | 128 | 74.6 | 83.6 | 89.9 | 94.0 | 85.4 | 91.6 | 94.8 | 97.1 | 82.7 | 92.6 | 96.7 | 98.7 | 90.9 | 97.9 | 98.6 | 98.9 |
| HIER-Poinc-DINO | 128 | 78.5 | 86.7 | 91.5 | - | 88.4 | **93.3** | 95.9 | - | 84.9 | 94.2 | 97.5 | - | 92.6 | **98.4** | 98.9 | - |
| **HIER-Klein-DINO** | 128 | 78.6 | 86.3 | 91.7 | 95.2 | **88.7** | **93.3** | **96.0** | 97.9 | 84.7 | 93.8 | 97.3 | 98.9 | 92.1 | 98.3 | 98.8 | 99.0 |
| HIER-Poinc-ViT-S | 128 | **84.2** | **90.1** | 93.7 | - | 86.4 | 91.9 | 95.1 | - | **85.6** | 94.6 | 97.8 | - | **92.7** | 98.4 | 98.9 | - |
| **HIER-Klein-ViT-S** | 128 | 83.8 | 90.0 | **94.1** | 96.3 | 85.4 | 91.5 | 95.2 | 97.2 | 85.3 | 94.4 | 97.7 | **99.0** | 92.6 | 98.4 | 98.9 | 99.1 |
| HIER-Poinc-DeiT-S | 384 | 78.7 | 86.8 | 92.0 | - | 88.9 | 93.9 | 96.6 | - | 83.0 | 93.1 | 97.2 | - | 90.6 | 98.1 | 98.6 | - |
| **HIER-Klein-DeiT-S** | 384 | 77.8 | 86.4 | 91.8 | 94.7 | 89.0 | 93.9 | 96.3 | 97.9 | 83.5 | 93.4 | 97.3 | 99.0 | 91.6 | 98.2 | 98.8 | 99.1 |
| HIER-Poinc-DINO | 384 | 81.1 | 88.2 | 93.3 | - | 91.3 | 95.2 | 97.1 | - | 85.7 | 94.6 | 97.8 | - | 92.5 | **98.6** | 99.0 | - |
| **HIER-Klein-DINO** | 384 | 81.2 | 88.7 | 92.8 | 95.9 | **91.5** | **95.3** | **97.3** | 98.5 | 85.9 | 94.8 | 97.9 | 99.2 | **92.9** | **98.6** | **99.1** | **99.3** |
| HIER-Poinc-ViT-S | 384 | **85.7** | **91.3** | 94.4 | - | 88.3 | 93.2 | 96.1 | - | 86.1 | 95.0 | 98.0 | - | 92.8 | 98.4 | 99.0 | - |
| **HIER-Klein-ViT-S** | 384 | 85.5 | 91.1 | **94.7** | 96.5 | 88.6 | 93.5 | 96.3 | 98.1 | **86.5** | 95.2 | 98.1 | **99.3** | 92.9 | 98.5 | **99.1** | 99.2 |

evaluating its performance with three different pre-trained checkpoints: the baseline ViT-S, DeiT-S, and DINO. During all fine-tuning stages, the patch embedding layer of the ViT backbone is frozen to preserve its fundamental feature extraction capabilities. Crucially, we rigorously tune the feature clipping radius $r$ for both Klein and Poincaré baselines independently to ensure optimal performance for each geometry.

### 4.3.1. IMPLEMENTATION WITH PCE LOSS

When training with the pairwise cross-entropy (PCE) loss Equation (7), the key hyperparameters are set as follows: the manifold curvature $c$ is fixed at 0.1, the loss temperature $\tau$ is 0.5, and the feature clipping radius $r$ is 4.8 for the CUB dataset and 4.9 for the others. For optimization, a weight decay of 0.1 is used, with a learning rate of 3e-5 for the DINO backbone and 5e-5 for ViT-S and DeiT-S. We use a batch size of 900 for CUB, SOP, and In-Shop. For the Cars dataset, which has 98 training classes, the batch size is 882. We sample 9 instances per class for CUB and Cars datasets, and 2 instances per class for SOP and In-Shop datasets. The models are trained for 50, 500, 200, and 400 epochs in CUB, Cars, SOP, and In-Shop, respectively, with gradient clipping applied at a global norm of 3. For evaluation, we utilize the cosine distance for the 384-dimensional Euclidean features and Einstein gyrovector distance $D_{gyr}^{\mathbb{K}}$ for the 128-dimensional hyperbolic embeddings.

### 4.3.2. IMPLEMENTATION WITH HIER LOSS

For our implementation with the HIER loss Equation (9), the central modification involves replacing the original Poincaré model with our Klein model. To ensure a fair and targeted comparison, we strictly adhere to all hyperparameter configurations from the original HIER publication (Kim et al., 2023), including its composite loss function, the internal parameters of its proxy-anchor loss $\mathcal{L}_{\mathrm{ML}}$, the weight $\lambda$, and the optimizer settings and learning rates.

### 4.4. Results and Analysis

As summarized in Table 1, the proposed deep metric learning framework based on the Klein model demonstrates highly competitive performance across all fine-grained benchmarks. The Klein model exhibits particularly significant advantages over the traditional Poincaré model on the Cars and In-shop datasets. For instance, in the 128-dimensional embedding space, Klein-DINO achieves a Recall@1 of 86.9%, representing a 3.4% absolute improvement over its Poincaré counterpart on the Cars dataset. Similarly, Klein-ViT-S reaches a Recall@1 of 93.5% on the In-shop dataset, surpassing both Poincaré and spherical baselines. These results provide empirical evidence that the projective geometry of the Klein model offers stronger

class discriminability for fine-grained classification tasks compared to conformal model.

To further investigate the impact of the Klein model on embedding quality, we compare the 128-dimensional intrinsic distance performance with the 384-dimensional original encoder features. Experimental results indicate that the Klein model maintains a leading or competitive position in both evaluation stages. Notably, the backbone features trained under the guidance of the Klein model outperform those trained with the Poincaré model in the 384-dimensional evaluation. This observation aligns with our theoretical derivation in Section 3.4, where the structurally more moderate metric expansion of the Klein model imposes less aggressive scaling on the backpropagated gradients. This enhanced stability optimizes the final geometric embeddings and improves the ability of the upstream encoder to extract discriminative features through backpropagation, thereby mitigating the gradient instability issues commonly encountered in hyperbolic training.

The validation across three distinct vision Transformer architectures including DeiT-S, DINO, and ViT-S confirms that the advantages of the Klein model are architecture-agnostic. The Klein model consistently enhances classification accuracy regardless of whether the backbone is self-supervised (DINO) or supervised (DeiT). In large-scale classification tasks such as In-shop, the Klein-based methods demonstrate exceptionally high precision in the Recall@1 metric. This suggests that the straight-line geodesic property is more effective at maintaining intra-class compactness within large-scale, high-dimensional feature distributions.

Table 2 illustrates the performance of the Klein model when applied to the hierarchical regularization (HIER) loss. Comparing this with the original Poincaré-based HIER framework reveals a compelling geometric trade-off. HIER-Klein demonstrates a clear advantage on the Cars and In-shop datasets, with the Cars-DINO architecture achieving Recall@1 values of 88.7% and 91.5% at different stages. These results indicate that the straight-line geodesics of the Klein model provide more precise hierarchical path alignment when processing inter-class structures with distinct visual features.

For datasets with high inter-class similarity and significant hierarchical depth such as CUB and SOP, HIER-Klein remains competitive with HIER-Poinc. Although HIER-Poinc maintains a marginal lead in certain Recall@1 metrics, HIER-Klein demonstrates stronger discriminative power in higher-order Recall metrics. For example, in the CUB-ViT-S 128D experiment, the Klein model improves Recall@4 from 93.7% to 94.1%. This phenomenon suggests that the non-conformal projective nature of the Klein model is more effective at reducing structural distortion within large-radius neighborhoods, thereby maintaining superior geometric con-

*Table 3.* Hyperparameter sensitivity analysis on In-Shop dataset using ViT-S backbone. An ablation study investigating the impact of key hyperparameter choices on the performance of the Klein model. We analyze the sensitivity to the hyperbolic curvature parameter ($c$) and the dimension of the embedding head.

| Parameters | Encoder (384) | Head (128) |
|---|---|---|
| Default | 93.0 | 93.5 |
| $c = 0.01$ | 80.5 | 78.0 |
| $c = 0.05$ | 90.0 | 90.3 |
| $c = 0.3$ | 89.5 | 89.5 |
| $c = 0.5$ | 88.3 | 88.2 |
| Head dim. 16 | 88.5 | 84.2 |
| Head dim. 32 | 90.0 | 89.6 |
| Head dim. 64 | 92.1 | 92.3 |

*Table 4.* Patch size analysis on Cars dataset using DINO backbone. We compare the standard 16×16 patch size against a finer 8×8 patch size across two embedding dimensions (128 and 384).

| Method | Dim | Recall@$K$ | | | |
|---|---|---|---|---|---|
| | | 1 | 2 | 4 | 8 |
| Klein-DINO 16×16 | 128 | 86.9 | 92.7 | 95.9 | 97.3 |
| Klein-DINO 8×8 | 128 | 91.2 | 95.2 | 97.3 | 98.3 |
| Klein-DINO 16×16 | 384 | 89.2 | 94.3 | 96.7 | 98.2 |
| Klein-DINO 8×8 | 384 | 93.6 | 96.5 | 98.1 | 98.9 |

sistency across the overall retrieval distribution when handling densely packed manifold embeddings.

### 4.5. Ablation Study

**Manifold Curvature** $c$**.** As shown in Table 3, our results show that the performance of the Klein model exhibits a significant sensitivity to the curvature parameter $c$. However, we posit that this apparent sensitivity is less an indication of a fixed optimal range for the curvature $c$ itself, and more a manifestation of the interdependent relationship between $c$ and the Euclidean feature clipping radius $r$. In this hyperparameter exploration, to simplify the search space, we adopted a strategy of keeping $\tanh(\sqrt{c} \times r)$ constant in order to maintain a roughly uniform hyperbolic embedding range across different curvatures. This strategy artificially established a strongly coupled inverse relationship between $c$ and $r$. Consequently, the performance degradation observed at low values of $c$ may not stem from the flatness of the space itself being suboptimal, but rather from our constraint forcing an excessively large clipping radius $r$, thereby inducing instability or information distortion during the mapping process. This suggests that the key challenge may not lie in finding an independent "optimal $c$ value" but rather in uncovering and leveraging the correct synergistic relationship between $c$ and $r$. For further investigation, please refer to Appendix B.1.

**Embedding Dimensionality.** Experimental results in Table 3 show a positive correlation between embedding dimensionality and model performance, where a reduction in the output dimension leads to a corresponding decrease in Recall. Nevertheless, the model maintains strong discriminative power even at lower dimensions. This is particularly noteworthy in our most challenging test data, which contains 3,985 classes, and confirms the ability of our embedding method to guide the encoder in learning discriminative features. This indicates that the Klein mode can compress

high-dimensional visual information into compact embeddings while preserving essential semantic features, thereby demonstrating excellent robustness and generalization potential even under significant dimensional constraints.

**Patch Size.** The vision Transformer (ViT) architecture operates not on individual pixels but by deconstructing the image into a sequence. The default patch size in a standard ViT is typically 16×16 pixels. However, prior research has demonstrated that employing a smaller patch size can yield substantial performance gains. Although reducing the patch size does not alter the total number of encoder parameters, it quadruples the length of the input sequence. This requires the encoder to process a much longer sequence, granting it the potential to capture more fine-grained visual details and learn more intricate inter-patch dependencies. To validate the effectiveness of this strategy on our task, we present a set of results using an 8×8 patch size in Table 4. Compared to the default 16×16 configuration, this setting yields a significant performance improvement of 4.3%. It is noted that in this ablation study we follow the same training protocol described in Subsection 4.3.1, with the batch size adjusted to 108 to accommodate the increased computational requirements.

## 5. Conclusion

This paper reinvestigates the value of the Klein model in metric learning by addressing the geometric misalignment between conformal hyperbolic model and the linear projection operations inherent in modern neural architectures. By formalizing a computational framework based on Einstein gyrovector space, we propose a consistent metric learning method. Leveraging its straight-line geodesics, the Klein model naturally aligns with the linear output spaces of modern backbones, achieving highly competitive performance without increasing model complexity. Our findings suggest a fundamental insight: when designing non-Euclidean metric learning frameworks, the alignment between the manifold's geometric properties and the encoder's inductive bias is a more critical factor than merely choosing a negative curvature space.

**Limitations and Future Work.** While we have demonstrated the geometric advantages of the Klein model, the current framework relies on a fixed curvature parameter, which limits its ability to fully exploit the underlying curvature distribution of complex data. Furthermore, while we have leveraged the alignment between the Klein model's Euclidean straight-line geodesics and the linear projections of modern encoders, this property opens a broader possibility: many mature loss functions designed for Euclidean spaces could potentially be adapted to the Klein model with minimal modification. Future work will focus on two directions: first, exploring learnable curvature mechanics to verify their potential in mapping heterogeneous geometric spaces; and second, designing generalized geometric operators and loss functions that exploit the Klein model's linearity to further bridge the operational gap between Euclidean deep learning and hyperbolic geometry.

## Acknowledgements

This work was supported by the National Natural Science Foundation of China (62476020, U25A20446).

## Impact Statement

This paper presents work whose goal is to advance the field of machine learning. There are many potential societal consequences of our work, none which we feel must be specifically highlighted here. This paper contributes to the fundamental understanding of hyperbolic geometry in metric learning by introducing the Klein model. While our work primarily focuses on the theoretical and algorithmic advancement of fine-grained visual recognition, the improved classification accuracy could benefit downstream applications such as biodiversity conservation (e.g., species identification) and large-scale digital asset management. We do not foresee any specific negative ethical or societal consequences resulting from this research.

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

# A. Mathematical Derivations

### A.1. Proving the Equivalence of Two Distance Forms in the Klein Model

For clarity of the subsequent derivation, we first simplify the square of the Einstein addition as

$$
\begin{aligned}
\|-\mathbf{x} \oplus_{\mathrm{E}} \mathbf{y}\|^2 &= \frac{1}{(1-c(\mathbf{x}^\mathsf{T}\mathbf{y}))^2} \left\| -\mathbf{x} + \frac{1}{\gamma_\mathbf{x}}\mathbf{y} + c\frac{\gamma_\mathbf{x}}{1+\gamma_\mathbf{x}}(\mathbf{x}^\mathsf{T}\mathbf{y})\mathbf{x} \right\|^2 \\
&= \frac{1}{(1-c(\mathbf{x}^\mathsf{T}\mathbf{y}))^2} \left( \left(1 - c\frac{\gamma_\mathbf{x}}{1+\gamma_\mathbf{x}}\mathbf{x}^\top\mathbf{y}\right)^2 \|\mathbf{x}\|^2 + \frac{1}{\gamma_\mathbf{x}^2}\|\mathbf{y}\|^2 - \frac{2}{\gamma_\mathbf{x}}\left(1 - c\frac{\gamma_\mathbf{x}}{1+\gamma_\mathbf{x}}\mathbf{x}^\top\mathbf{y}\right)\mathbf{x}^\top\mathbf{y}\right) \\
&= \frac{1}{(1-c(\mathbf{x}^\mathsf{T}\mathbf{y}))^2} \left( \|\mathbf{x}\|^2 + \frac{c^2(\mathbf{x}^\mathsf{T}\mathbf{y})^2\|\mathbf{x}\|^2}{2 - c\|\mathbf{x}\|^2 + 2\sqrt{1-c\|\mathbf{x}\|^2}} - \frac{2c\mathbf{x}^\mathsf{T}\mathbf{y}\|\mathbf{x}\|^2}{1 + \sqrt{1-c\|\mathbf{x}\|^2}} + \|\mathbf{y}\|^2 - c\|\mathbf{x}\|^2\|\mathbf{y}\|^2 \right. \\
&\quad \left. -2\sqrt{1-c\|\mathbf{x}\|^2}\mathbf{x}^\mathsf{T}\mathbf{y} + \frac{2c\sqrt{1-c\|\mathbf{x}\|^2}}{1 + \sqrt{1-c\|\mathbf{x}\|^2}}(\mathbf{x}^\mathsf{T}\mathbf{y})^2 \right) \\
&= \frac{\|\mathbf{x} - \mathbf{y}\|^2 - c\|\mathbf{x}\|^2\|\mathbf{y}\|^2 + c(\mathbf{x}^\top\mathbf{y})^2}{(1-c\mathbf{x}^\mathsf{T}\mathbf{y})^2}.
\end{aligned}
\tag{12}
$$

Subsequently, we address the gyrovector distance of the Klein model and the standard geodesic distance separately. The Einstein gyrovector distance is given by

$$
\begin{aligned}
D_{\mathrm{gyr}}^{\mathbb{K}}(\mathbf{x},\mathbf{y}) &= \frac{1}{\sqrt{c}}\operatorname{arctanh}(\sqrt{c}\|-\mathbf{x} \oplus_E \mathbf{y}\|) \\
&= \frac{1}{2\sqrt{c}}\ln\frac{\left(1 + \sqrt{c}\|-\mathbf{x} \oplus_E \mathbf{y}\|\right)^2}{1 - c\|\mathbf{x} \oplus_E \mathbf{y}\|^2} \\
&= \frac{1}{2\sqrt{c}}\ln\frac{\left(1 - c\mathbf{x}^\mathsf{T}\mathbf{y} + \sqrt{c}\sqrt{\|\mathbf{x}-\mathbf{y}\|^2 - c\|\mathbf{x}\|^2\|\mathbf{y}\|^2 + c(\mathbf{x}^\mathsf{T}\mathbf{y})^2}\right)^2}{(1 - c\|\mathbf{x}\|^2)(1 - c\|\mathbf{y}\|^2)}.
\end{aligned}
\tag{13}
$$

The Klein distance is given by

$$
\begin{aligned}
D^{\mathbb{K}}(\mathbf{x},\mathbf{y}) &= \frac{1}{\sqrt{c}}\operatorname{arcosh}\left(\frac{1 - c\mathbf{x}^\mathsf{T}\mathbf{y}}{\sqrt{1-c\|\mathbf{x}\|^2}\sqrt{1-c\|\mathbf{y}\|^2}}\right) \\
&= \frac{1}{\sqrt{c}}\ln\left(\frac{1 - c\mathbf{x}^\mathsf{T}\mathbf{y}}{\sqrt{1-c\|\mathbf{x}\|^2}\sqrt{1-c\|\mathbf{y}\|^2}} + \sqrt{\left(\frac{1 - c\mathbf{x}^\mathsf{T}\mathbf{y}}{\sqrt{1-c\|\mathbf{x}\|^2}\sqrt{1-c\|\mathbf{y}\|^2}}\right)^2 - 1}\right) \\
&= \frac{1}{\sqrt{c}}\ln\left(\frac{1 - c\mathbf{x}^\mathsf{T}\mathbf{y}}{\sqrt{1-c\|\mathbf{x}\|^2}\sqrt{1-c\|\mathbf{y}\|^2}} + \frac{\sqrt{c(\|\mathbf{x}-\mathbf{y}\|^2 - c\|\mathbf{x}\|^2\|\mathbf{y}\|^2 + c(\mathbf{x}^\mathsf{T}\mathbf{y})^2)}}{\sqrt{1-c\|\mathbf{x}\|^2}\sqrt{1-c\|\mathbf{y}\|^2}}\right).
\end{aligned}
\tag{14}
$$

Consequently, we derive

$$
D^{\mathbb{K}}(\mathbf{x},\mathbf{y}) = D_{\mathrm{gyr}}^{\mathbb{K}}(\mathbf{x},\mathbf{y}).
\tag{15}
$$

### A.2. Illustrating the Training Stability of Equation (4) vs. Equation (3) with a Simple Case

In practical training, positive samples are pulled together and negative samples are pushed away. Nevertheless, as the distance between sample pairs approaches zero, leveraging Equation (3) yields gradients that lead to severe numerical instability. The gradient computation under the PyTorch framework strictly follows the chain rule by multiplying step by step, rather than using the simplified result. For example, in the simple case $\mathbf{x} = \mathbf{0}$, letting $r = |\mathbf{y}| \to 0$, then the inner term of the function for Equation (3) is $z(r) = \frac{1}{\sqrt{1-cr^2}}$, and the total distance function is $D^{\mathbb{K}}(r) = \frac{1}{\sqrt{c}}\operatorname{arcosh}(z(r))$. In the computational graph of PyTorch, backpropagation is calculated independently according to the chain rule. The derivative of the outer function with respect to $z$ is: $\frac{dD^{\mathbb{K}}}{dz} = \frac{1}{\sqrt{c}} \cdot \frac{1}{\sqrt{z^2-1}} = \frac{\sqrt{1-cr^2}}{cr}$. When $r \to 0$, this intermediate gradient

*Table 5.* Performance comparison (Recall@$K$) of different distance functions across various model architectures and feature dimensions on the Cars dataset.

| Distance Function | Model Architecture | Recall@1 (%) | | Recall@2 (%) | | Recall@4 (%) | | Recall@8 (%) | |
|---|---|---|---|---|---|---|---|---|---|
| | | 128-D | 384-D | 128-D | 384-D | 128-D | 384-D | 128-D | 384-D |
| Equation (3) | ViT-S | 81.11 | 84.05 | 88.12 | 90.57 | 92.62 | 94.44 | 95.82 | 96.85 |
| **Equation (4)** | **ViT-S** | **83.38** | **86.46** | **90.42** | **92.40** | **94.45** | **95.84** | **97.25** | **97.98** |
| Equation (3) | DINO | 81.98 | 87.22 | 88.34 | 92.25 | 92.66 | 95.35 | 95.50 | 97.37 |
| **Equation (4)** | **DINO** | **86.89** | **89.24** | **92.66** | **94.26** | **95.88** | **96.65** | **97.33** | **98.24** |
| Equation (3) | DeiT-S | 79.26 | 83.79 | 86.72 | 90.43 | 91.61 | 94.06 | 95.15 | 96.54 |
| **Equation (4)** | **DeiT-S** | **82.25** | **85.72** | **89.79** | **92.02** | **94.07** | **95.54** | **96.59** | **97.68** |

will be very large, which will produce "NaN" during training. The derivative of the inner function with respect to $r$ is: $\frac{dz}{dr} = -\frac{1}{2}(1 - cr^2)^{-3/2} \cdot (-2cr) = \frac{cr}{(1-cr^2)^{3/2}}$. Therefore, while $r$ and $\frac{1}{r}$ in $\frac{dD^{\mathbb{K}}}{dz} \times \frac{dz}{dr}$ cancel out algebraically, in the computer's floating-point system, the computational graph overflows and crashes at the first step because a huge gradient has already been calculated. In contrast, the intermediate derivatives of Equation (4) are very stable. Its inner term is $u(r) = \sqrt{c}r$, and the total distance function is $D^{\mathbb{K}}_{\text{gyr}}(r) = \frac{1}{\sqrt{c}}\text{artanh}(u(r))$. Next, we calculate the intermediate derivative for backpropagation: $\frac{dD^{\mathbb{K}}_{\text{gyr}}}{du} = \frac{1}{\sqrt{c}}\frac{1}{1-u^2} = \frac{1}{\sqrt{c}(1-cr^2)}$, the value of this gradient when $r \to 0$ is $\frac{1}{\sqrt{c}}$, and $\frac{du}{dr} = \sqrt{c}$. Therefore, under the PyTorch deep learning framework, Equation (4) has better numerical stability than Equation (3). To corroborate our derivation, we compare the classification performance of these two distance formulas on the Cars dataset, and the results are as follows:

The experimental results show that the performance of Equation (4) is better than that of Equation (3), being 2%-5% higher on every item in Recall@1.

### A.3. Deriving the Inverse of the Riemannian Gradient (Scaling Factor) from the Line Element

The forms of the line element and the scaling factor are presented in the paper. We replace $\mathbf{x}$ with $\mathbf{z}$, $\mathbf{dx}$ with $\mathbf{dz}$, and utilize the definition of the line element in Riemannian geometry $ds^2 = \mathbf{dz}^{\top}\mathbf{G}(\mathbf{z})\mathbf{dz}$.. Write $\|\mathbf{dz}\|^2$ as $\mathbf{dz}^{\top}\mathbf{I}_n\mathbf{dz}$, and write $(\mathbf{z}^{\top}\mathbf{dz})^2$ as $\mathbf{dz}^{\top}(\mathbf{z}\mathbf{z}^{\top})\mathbf{dz}$. The line element becomes

$$ds^2_{\mathbb{K}} = \mathbf{dz}^{\top}\left(\frac{1}{1 - c\|\mathbf{z}\|^2}\mathbf{I}_n + \frac{c}{(1 - c\|\mathbf{z}\|^2)^2}\mathbf{z}\mathbf{z}^{\top}\right)\mathbf{dz}. \tag{16}$$

Therefore, the metric tensor $\mathbf{G}$ of the Klein model is given by

$$\mathbf{G}^{\mathbb{K}}(\mathbf{z}) = \frac{1}{1 - c\|\mathbf{z}\|^2}\mathbf{I}_n + \frac{c}{(1 - c\|\mathbf{z}\|^2)^2}\mathbf{z}\mathbf{z}^{\top}. \tag{17}$$

Set $\mathbf{G}^{\mathbb{K}}(\mathbf{z}) = a\mathbf{I}_n + b\mathbf{z}\mathbf{z}^{\top}$, compute the inverse using the Sherman-Morrison formula

$$\mathbf{G}^{\mathbb{K}}(\mathbf{z})^{-1} = \frac{1}{a}\mathbf{I}_n - \frac{b}{a(a + b\|\mathbf{z}\|^2)}\mathbf{z}\mathbf{z}^{\top}. \tag{18}$$

Substitute $a = \frac{1}{1-c\|\mathbf{z}\|^2}$ and $b = \frac{c}{(1-c\|\mathbf{z}\|^2)^2}$ into the equation, yielding the expression in Equation (10).

## B. More Experimental Analysis

### B.1. Analysis of Curvature Parameter and Feature Clipping Size

Our experimental investigation is predicated on the assumption that $\tanh(\sqrt{c} \times r)$ remains a fixed value. This strategy is designed to preserve the ratio between the vector length and the model radius after feature clipping and exponential mapping into the Klein model. However, as shown in Figure 3(a) and Figure 3(b), when the curvature parameter $c$ is small, the model fails to achieve optimal performance. Combined with our experimental observations, we posit that this is due to the significant increase in the Klein model's radius when $c$ decreases. This expansion leads to the generation of overly extreme

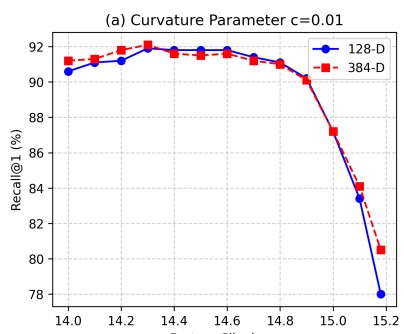 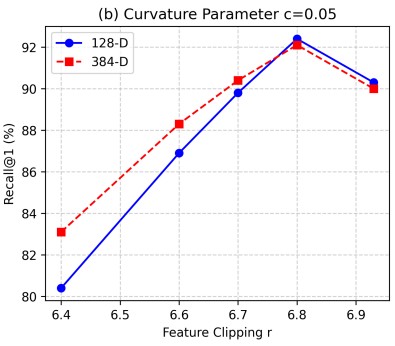 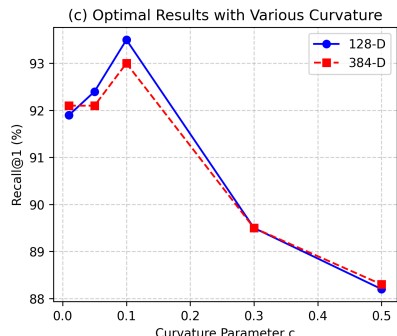

*Figure 3.* Relationship between curvature and feature clipping. (a) and (b) present the performance trends across different feature clipping radii for curvature parameters $c = 0.01$ and $c = 0.05$, respectively, where $r = 15.18$ and $r = 6.93$ represent the theoretical geometric default values for these two cases. We observe that a moderate reduction in the feature clipping radius leads to an improvement in performance. (c) provides the results corresponding to the optimal feature clipping radius for various curvature parameters, demonstrating that the model performance is relatively robust to changes in curvature.

metric scaling during training, which induces numerical instability to distinguish positive and negative samples, resulting in the excessive distortion of the embedding manifold. This high degree of geometric distortion causes overfitting and the degradation of the learned topological structure. Conversely, a slightly more conservative clipping acts as an effective geometric regularizer. By preventing the embeddings from entering the most sensitive region near the boundary, it limits the exposure to metric inflation, facilitating a more stable optimization process that preserves the structure required for strong generalization.

When the curvature parameter $c$ is increased, optimal performance is achieved while maintaining $\tanh(\sqrt{c} \times r)$ as a constant. However, since the model radius $\frac{1}{\sqrt{c}}$ simultaneously decreases, the local expansion rate of the hyperbolic space becomes highly sensitive to radial changes. Consequently, even a minute adjustment to the feature clipping range $r$ can lead to extreme volatility in performance.

Based on the analysis of these two cases, we conclude that the optimal choice of the feature clipping radius $r$ is a delicate balance between geometric proportionality and optimization stability. When the curvature $c$ is small, the primary challenge is the excessive spatial expansion caused by the large disk radius, where a theoretically permissible $r$ leads to overfitting. Conversely, when $c$ is large, the reduced disk radius amplifies the local geometric sensitivity, making the performance highly volatile to minute adjustments in $r$. This bidirectional volatility underscores the critical role of $r$ as an empirical regulator essential for maintaining a stable optimization manifold across the full range of hyperbolic curvatures.

### B.2. Comparing the Structural Distortion between Hyperbolic Models and Euclidean Encoders

To evaluate which hyperbolic space is more compatible with modern backbones, we randomly generate 100,000 Euclidean feature triplets with varying degrees of linearity. After applying the exact feature clipping and exponential maps for both models, we calculate the geodesic deviation error (GDE), which measures how much a straight Euclidean path is bent into a curve on the manifold. Specifically, we first select triplets of nearly collinear points $(z_1, z_2, z_3)$ from the Euclidean feature space, where $z_2$ lies approximately on the straight segment connecting $z_1$ and $z_3$. $GDE$ is then mathematically defined as: $\text{GDE}(z_1, z_2, z_3) = \frac{d(z_1,z_2)+d(z_2,z_3)}{d(z_1,z_3)} - 1$, where $d(\cdot, \cdot)$ is the distance function in the corresponding space (i.e., Euclidean norm or hyperbolic distance). A GDE value of $0$ indicates perfect collinearity. A larger positive value signifies a greater deviation from this linear structure, indicating that the path is more bent. The probability of the Klein model preserving the linear structure better than the Poincaré model ($\text{GDE}_{\mathbb{K}} < \text{GDE}_{\mathbb{P}}$) is summarized in Table 6.

As the results show, when the Euclidean features are highly collinear ($\text{GDE}_{Euc} < 0.01$), the Klein model perfectly preserves this structural topology 97.73% of the time. Conversely, Poincaré's curved arcs force these paths to bend, causing severe structural misalignment.

In order to verify whether GDE has a similar performance on real encoder features, we conduct an experimental verification using the DINO architecture on the CUB dataset. We connect the classification heads of the Klein model and the Poincaré model to DINO and train them respectively. Because different classification heads will affect the parameter changes of

*Table 6.* Comparison of Geodesic Deviation Error (GDE) on randomly generated Euclidean feature triplets.

| Euclidean Linearity | Klein Winning Rate ($\mathrm{GDE}_\mathbb{K} < \mathrm{GDE}_\mathbb{P}$) |
|---|---|
| $\mathrm{GDE}_{Euc} < 0.01$ | **97.73%** |
| $\mathrm{GDE}_{Euc} < 0.05$ | **95.73%** |
| $\mathrm{GDE}_{Euc} < 0.1$ | **94.87%** |
| $\mathrm{GDE}_{Euc} < 0.2$ | **93.59%** |

*Table 7.* Comparison of Geodesic Deviation Error (GDE) on real encoder feature triplets extracted from the CUB dataset using the DINO architecture.

| Metric space | $\mathrm{GDE}_{Euc}$ | $\frac{\mathrm{GDE}_{hyp}}{\mathrm{GDE}_{Euc}}$ |
|---|---|---|
| Klein | $< 0.2$ | 1.0021 |
| Poincaré | $< 0.2$ | 1.1398 |
| Klein | $< 0.1$ | 1.0269 |
| Poincaré | $< 0.1$ | 1.0536 |
| Klein | $< 0.05$ | 1.0271 |
| Poincaré | $< 0.05$ | 1.0543 |

the backbone model during training, we choose $\frac{\mathrm{GDE}_{hyp}}{\mathrm{GDE}_{Euc}}$ ($\mathrm{GDE}_{hyp}$ refers to $\mathrm{GDE}_\mathbb{K}$ or $\mathrm{GDE}_\mathbb{P}$) as the evaluation metric for comparison. The experimental results are shown in Table 7.

The experimental results show that even on real encoder features, the Klein model is still able to better preserve the linear structural topology than the Poincaré model.

