# OpenReview forum: "Klein Hyperbolic Metric Learning"
_ICML.cc/2026/Conference — ICML 2026 regular_

### Official Review · Reviewer_f1QV · 2026-02-26

**Soundness:** 3
**Presentation:** 2
**Significance:** 2
**Originality:** 2
**Overall Recommendation:** 3
**Confidence:** 4

**Summary:**

The paper proposes hyperbolic metric learning in the Klein model instead of the usual Poincaré model, mapping backbone features into a Klein hyperbolic ball and training them with a Klein/gyrovector distance-based contrastive objective, plus a proxy-based hierarchical regularizer and a boundary-clipping stabilization to handle numerical issues near the ball boundary.

**Compliance With Llm Reviewing Policy:**

Affirmed.

**Final Justification:**

My main concerns were largely addressed during the rebuttal, especially through clearer motivation and better explanation of the design choices. Although some claims should still be stated more cautiously in the revised manuscript, I find the contribution meaningful overall.

**Key Questions For Authors:**

For detailed issues, please refer to the Weaknesses section above.

**Limitations:**

yes

**Strengths And Weaknesses:**

Strengths
It studies an important design choice for hyperbolic DML and provides a practical, plug-and-play training recipe with comparisons to baselines and useful sensitivity analysis on curvature/stabilization.

Weaknesses
Major：
1. The paper attributes Poincaré optimization difficulty mainly to curved geodesics, but Poincaré optimization is typically performed via log/exp maps in local (Euclidean) tangent spaces, so the causal link between “curved geodesics” and optimization hardness is not rigorously established and should be clarified with a more precise explanation.

2. Since the Klein model is not conformal (angles are not preserved), the paper should discuss whether and how this choice may introduce additional distortion (e.g., for noisy features) and how such distortion affects metric learning performance.

3. Eq. (5) resembles the exponential map commonly used for the Poincaré ball at the origin, so the manuscript must explicitly specify the coordinate system of each embedding (Klein vs. Poincaré) and whether/where a Poincaré↔Klein conversion is applied; otherwise the mathematical pipeline is difficult to verify and reproduce.

4. The curvature-sensitivity ablation is informative but currently narrow, and should be evaluated across additional datasets and/or backbones to support the generality of the experimental conclusions.

Minor：
1. Notation is occasionally inconsistent or underspecified.
2. Experimental tables report single numbers without variance, making it difficult to assess the stability of the reported gains.

---

> ### Author Rebuttal · Authors · 2026-03-31
>
> We thank you for your insightful comments, and we address your concerns below.
>
> **Q1:** The paper attributes Poincaré optimization difficulty mainly to curved geodesics, but Poincaré optimization is typically performed via log/exp maps in local tangent spaces, so the causal link between “curved geodesics” and optimization hardness is not rigorously established and should be clarified with a more precise explanation.
>
> **Response:** Many thanks for your comment. We appreciate your rigorous correction. We entirely agree that during Riemannian SGD, parameter updates are indeed projected back to the local Euclidean tangent space via log/exp maps. Specifically, we acknowledge that our work conflates the representational limitation with the numerical optimization process. We will clarify this in the revised version by providing a more precise explanation.
>
> **Q2:** Since the Klein model is not conformal, the paper should discuss whether and how this choice may introduce additional distortion and how such distortion affects metric learning performance.
>
> **Response:** Many thanks for your comment. We fully understand your concern regarding non-conformal properties. However, in the deep metric learning framework used in our study, the core optimization criterion relies strictly on the relative distances between samples, not the angles between them. Therefore, in our study, adopting the Klein model does not introduce harmful distortion.
>
> **Q3:** Eq. (5) resembles the exponential map commonly used for the Poincaré ball at the origin, so the manuscript must explicitly specify the coordinate system of each embedding and whether/where a Poincaré↔Klein conversion is applied.
>
> **Response:** Thank you for pointing out this. We explicitly confirm that there is no intermediate conversion between Poincaré and Klein coordinates in our mathematical pipeline. In differential geometry, because the Riemannian metric tensors of the Klein and Poincaré models coincide exactly at the origin, the analytical expressions for their exponential maps from the tangent space at the origin are algebraically identical. Eq. (5) is directly formulated as the origin exponential map for the Klein model. We treat the Euclidean backbone features directly as vectors in the origin tangent space $T_0\mathbb{K}_c^n$ and map them in a single step into the Klein manifold $\mathbb{K}_c^n$. We will explicitly denote the coordinate systems for each embedding in the revised version to eliminate any ambiguity.
>
> **Q4:** The curvature-sensitivity ablation is informative but currently narrow, and should be evaluated across additional datasets and/or backbones to support the generality of the experimental conclusions.
>
> **Response:** Many thanks. Your suggestion is very constructive. To prove the generality of the observed sensitivity patterns, we conducted an additional joint ablation study over curvature ($c$) and clipping radius ($r$) on the Cars dataset (using the DINO backbone). We found that while the specific magnitudes of sensitivity vary across different dataset/backbone combinations under the constraint $\text{tanh}(\sqrt{c}×r)=const$, the underlying coupling phenomenon strictly aligns with our theoretical analysis in Appendix B.1: small curvatures lead to massive manifold radii that cause underfitting, whereas large curvatures result in extreme sensitivity to the clipping radius. Thoroughly decoupling and adaptively learning these two parameters necessitates a specialized curvature optimization mechanism, which remains an independent, open challenge in hyperbolic geometry.
>
> | **Feature Clipping** ($c=0.01$)  | 14.2 | 14.4 | 14.6 | 14.8 | 15.0 | 15.18 |
> | :--- | :---: | :---: | :---: | :---: | :---: | :---: |
> | **Recall@1 (%) 128-D & 384-D** | 79.6 & 87.6 | 80.8 & 87.9 | 82.5 & 88.5 | 82.4 & 86.8 | 80.2 & 84.9 | 74.9 & 80.9 |
>
> | **Feature Clipping** ($c=0.05$)  | 6.6 | 6.7 | 6.8 | 6.93 |
> | :--- | :---: | :---: | :---: | :---: |
> | **Recall@1 (%) 128-D & 384-D** | 69.3 & 71.7 | 73.9 & 75.9 | 81.5 & 87.0 | 76.2 & 80.9 |
>
> | **Curvature($c$)**   | 0.01 | 0.05 | 0.1 | 0.3 | 0.5 |
> | :--- | :---: | :---: | :---: | :---: | :---: |
> | **Recall@1 (%) 128-D & 384-D** | 82.5 & 88.5 | 81.5 & 87.0 | 86.9 & 89.2 | 85.3 & 88.7 | 61.7 & 65.0 |
>
> **Q5:** Notation is occasionally inconsistent or underspecified.
>
> **Response:** Thank you for pointing out this. We will clarify them in the revised version.
>
> **Q6:** Experimental tables report single numbers without variance.
>
> **Response:** Many thanks for your suggestion. To verify the stability of our proposed method, we conducted three independent runs on the Cars dataset using the DINO architecture. The resulting Recall@1 for the two classification heads are 86.47±0.39% and 89.43±0.27%, respectively. Such minimal variance demonstrates that our reported performance gains are stable and statistically significant. We will include the variance in tables to show the stability of the reported gains in the revised version.

---

> > ### Author Rebuttal · Reviewer_f1QV · 2026-04-05
> >
> > Thank you for the rebuttal. I appreciate the authors’ effort and find the clarifications for Q3 and Q4 helpful. However, my core concern remains unresolved.
> >
> > In particular, for Q1, the rebuttal acknowledges that the original manuscript conflates representational limitations with numerical optimization difficulty, but it does not provide a rigorous replacement for the original causal argument. Since this point is central to the paper’s motivation and technical justification, I do not consider it resolved by a promise of clarification in the revision.
> >
> > I also remain unconvinced by the response to Q2. Therefore, I consider my concerns to be partially resolved or unresolved.

---

> > > ### Author Response · Authors · 2026-04-05
> > >
> > > **Q1:** In particular, for Q1, the rebuttal acknowledges that the original manuscript conflates representational limitations with numerical optimization difficulty, but it does not provide a rigorous replacement for the original causal argument. Since this point is central to the paper’s motivation and technical justification, I do not consider it resolved by a promise of clarification in the revision.
> > >
> > > **Response:** Many thanks for your rigorous comments for the motivation of the paper. We clarify once again that our motivation for selecting the Klein model is that the backbone model naturally tends to represent semantic transitions as straight-line interpolations in the latent space, and the straight-line geodesics of the Klein model are more suitable for modern backbone models. If the Poincaré model is adopted, because its geodesics are curved arcs, the loss function is actually forcing the network to align features along curved paths. This systematically distorts the straight-line semantic topology that the backbone network intends to maintain, forcing the network to produce unnatural geometric deformations in the latent space, which is exactly the fundamental causal mechanism leading to severe representational limitations of the model. We design experiments to prove our point. We randomly generate 100,000 Euclidean feature triplets with varying degrees of linearity. After applying the exact feature clipping and exponential maps for both models, we calculate the Geodesic Deviation Error (GDE), which measures how much a straight Euclidean path is bent into a curve on the manifold. The probability of the Klein model preserving the linear structure better than Poincaré model ($GDE_\mathbb{K}<GDE_\mathbb{P}$) is summarized in the table below:
> > >
> > > | Euclidean Linearity | $GDE_{Euc} < 0.01$ | $GDE_{Euc} < 0.05$ | $GDE_{Euc} < 0.10$ | $GDE_{Euc} < 0.20$ |
> > > | :--- | :--- | :--- | :--- | :--- |
> > > | **Klein Winning Rate ($GDE_{\mathbb{K}} < GDE_{\mathbb{P}}$)** | **97.73%** | **95.73%** | **94.87%** | **93.59%** |
> > >
> > > In order to verify whether GDE has a similar performance on real encoder features, we conduct an experimental verification using the DINO architecture on the CUB dataset. We connect the classification heads of the Klein model and the Poincaré model to DINO and train them respectively. Because different classification heads will affect the parameter changes of the backbone model during training, we choose $\frac{GDE_{hyp}}{GDE_{Euc}}$ ($GDE_{hyp}$ refers to $GDE_\mathbb{K}$ or $GDE_\mathbb{P}$, where the ratio closer to 1 is better) as the evaluation metric for comparison. The experimental results are as follows:
> > >
> > > | Metric space | $GDE_{Euc}$ | $\frac{GDE_{hyp}}{GDE_{Euc}}$ |
> > > | :--- | :--- | :--- |
> > > | Klein | <0.2 | 1.0021 |
> > > | Poincaré | <0.2 | 1.1398 |
> > > | Klein | <0.1 | 1.0269 |
> > > | Poincaré | <0.1 | 1.0536 |
> > > | Klein | <0.05 | 1.0271 |
> > > | Poincaré | <0.05 | 1.0543 |
> > >
> > > The experimental results prove that our claim that the Poincaré model has representational limitation and that the Klein model better solves this issue is valid. Based on the above content, we will modify the motivation and technical justification of the paper in the revised manuscript.
> > >
> > > **Q2:** I also remain unconvinced by the response to Q2.
> > >
> > > **Response:** We believe that the above experiment regarding GDE can also explain this problem. The Klein model is indeed non-conformal; it sacrifices angle-preservation, which inevitably introduces local angular distortion. However, for a backbone model that outputs Euclidean features, if the Poincaré model is used, although it keeps local angles unchanged, it severely bends these semantic paths that should be straight lines. As demonstrated in our experiment in Q1, the Poincaré model causes more structural distortion to the real feature paths. In contrast, although the Klein model tolerates local angular distortion, it trades it for global projectivity (preserving straight lines). The final experimental results show that for linear backbone networks, the optimization benefits brought by eliminating path-bending distortion exceed the harm caused by sacrificing local conformality. Therefore, adopting the Klein model does not introduce harmful distortion, but rather makes the most beneficial geometric trade-off for metric learning.
> > >
> > > Thank you again and hope this address your concerns.

---

### Official Review · Reviewer_b86z · 2026-03-01

**Soundness:** 4
**Presentation:** 3
**Significance:** 3
**Originality:** 3
**Overall Recommendation:** 5
**Confidence:** 5

**Summary:**

This paper studies hyperbolic metric learning in the Klein model, an alternative to the Poincaré model. The authors argue that the straight-line geodesics in the Klein model are better aligned with the linear transformations in modern neural network backbones. The method is evaluated on several fine-grained image retrieval benchmarks, where it demonstrates competitive or improved performance compared to Poincaré-based and hyperspherical baselines. The work positions the Klein model as a practical and effective geometric prior for hyperbolic metric learning.

**Compliance With Llm Reviewing Policy:**

Affirmed.

**Final Justification:**

I appreciate the authors’ detailed rebuttal and follow-up clarifications. Overall, my main concerns have been adequately addressed, and I have updated my score accordingly. In particular, the authors provided substantially clearer explanations regarding the numerical stability issue, the role of hyperspherical space in Section 3.3, the motivation for introducing hyperbolic geometry at the final stage, and the intended revisions to presentation details. While I still think the discussion related to customized backpropagation, as well as the contribution statement of Section 3.3, could be further refined in the final version, these no longer affect my overall assessment. Overall, I support acceptance.

**Key Questions For Authors:**

See above.

**Limitations:**

Yes

**Strengths And Weaknesses:**

## Strengths
- 1. The experimental evaluation is comprehensive, covering multiple benchmarks and settings.
- 2. The experimental section is clearly written and easy to follow.
- 3. The ablation studies provide a detailed analysis of the key components.
- 4. The introduction of the Klein model is well motivated.

## Weaknesses
- 1. The claimed numerical stability advantage of Eq. (4) over Eq. (3) is not substantiated, since Eq. (4) still relies on Einstein addition with small-denominator terms, e.g., $1 + c x^T y$, that may cause boundary-related amplification, $\mathrm{artanh}$ also becomes ill-conditioned as its argument approaches 1, and can yield NaNs under finite precision. In addition, any stabilization via norm clipping would equally apply to Eq. (3). The asserted advantage is unclear.
- 2. Section 3.3 suddenly introduces hyperspherical space, which is confusing and insufficiently motivated.
- 3. I am unclear about the contribution of Section 3.3, as it appears that the authors mainly replace the original Poincaré distance in the Hierarchical Regularization Loss with the Klein geodesic distance.
- 4. Since the backbone network operates in Euclidean space and only the final metric learning stage is in hyperbolic space, it would be useful to clarify why introducing hyperbolic geometry at the final stage is necessary and what additional benefits it brings beyond the Euclidean representation. I think a common introduction of hyperbolic geometry is the hierarchical structure in the data. Recent works have quantitatively analyzed hyperbolicity to support design choices. So, the authors need to justify this geometric assumption by conducting a quantitative analysis, e.g., by measuring hyperbolicity,
- 5. Presentation Suggestions
  - It may improve readability to add a horizontal separator between different backbones in Tables 1 and 2.
  - Reporting the mean performance across datasets could help readers more easily compare overall trends between methods.
  - It would be helpful to clarify in the table captions that the columns (e.g., 1, 2, 4, 8) correspond to Recall@K.

Overall, this is an interesting work. However, the above concerns need to be addressed. If the authors can satisfactorily resolve these issues, I would be willing to raise my score to Accept.

---

> ### Author Rebuttal · Authors · 2026-03-31
>
> We thank you for your insightful comments, and we address your concerns below.
>
> **Q1:** The claimed numerical stability advantage of Eq. (4) over Eq. (3) is not substantiated. In addition, any stabilization via norm clipping would equally apply to Eq. (3). The asserted advantage is unclear.
>
> **Response:** Many thanks for your comment. Your observation is highly astute: feature clipping does indeed prevent the denominator from reaching absolute zero at the manifold boundary. However, the fatal numerical instability of Eq. (3) does not occur at the boundary, but rather at the origin of the distance function (when x→y), the most frequent scenario in metric learning when pulling positive pairs together. From a calculus limit perspective, the gradient of the standard distance (Eq. 3) relies on the inverse hyperbolic cosine: $\frac{d}{dz}\text{arcosh}(z) = \frac{1}{\sqrt{z^2-1}}$. As positive samples converge ($x→y$), the internal term $z→1$. Consequently, the gradient approaches infinity. In contrast, our gyrovector distance (Eq. 4) relies on the inverse hyperbolic tangent: $\frac{d}{dz}\text{artanh}(z) = \frac{1}{1-z^2}$. As samples converge ($x→y$), the gyrovector difference vanishes $z→0$. The gradient gracefully evaluates to $1$. Thus, regardless of whether features are near the boundary or the origin, Eq. (4) guarantees a perfectly well-conditioned, smooth gradient, completely avoiding the singularity caused by clustered features. We will formally include this derivative proof in revised Appendix.
>
> **Q2:** Section 3.3 suddenly introduces hyperspherical space, which is confusing and insufficiently motivated.
>
> **Response:** Many thanks for your comment. The pioneering HIER framework [1] successfully combines hyperspherical space and hyperbolic space to leverage their complementary strengths, relying fundamentally on the conformal Poincaré model. In our study, we integrate the Klein model into the HIER framework, which utilizes both hyperspherical and Klein embedding spaces. Consequently, the hyperspherical space is introduced in Section 3.3 to maintain the structural integrity. We will add a brief clarification in the revised version.
>
> [1] Kim, S., Jeong, B., and Kwak, S. Hier: Metric learning beyond class labels via hierarchical regularization. In Proceedings of the IEEE/CVF Conference on Computer Vision and Pattern Recognition, pp. 19903-19912, 2023.
>
> **Q3:** I am unclear about the contribution of Section 3.3, as it appears that the authors mainly replace the original Poincaré distance in the Hierarchical Regularization Loss with the Klein geodesic distance.
>
> **Response:** Many thanks for your comment. We clarify that Section 3.3 is not merely a replacement of the distance function, but a system-level integration of the Klein model into the HIER framework. This process involves not only the metric transition but also the adaptation of manifold-specific operations, such as the exponential map and feature clipping. The contribution of this section lies in demonstrating the geometric compatibility of the Klein model within a multi-space paradigm.
>
> **Q4:** Clarify why introducing hyperbolic geometry at the final stage is necessary and what additional benefits it brings beyond the Euclidean representation. Justify this geometric assumption by conducting a quantitative analysis, e.g., by measuring hyperbolicity,
>
> **Response:** Many thanks for your comment. This is a fundamental and important question. The core necessity for introducing hyperbolic geometry lies in the implicit hierarchical structures inherent in natural fine-grained data (e.g., taxonomies of species). Regarding the quantitative analysis to justify this geometric assumption, prior foundational literature such as [2] has already extensively measured and rigorously verified that the fine-grained datasets used in our study possess extremely low Gromov $\delta$-hyperbolicity. This implies that the data itself intrinsically conforms to hyperbolic geometric assumption. To make our argumentation more rigorous, we will aggregate the quantitative hyperbolicity metric from existing literature for these datasets in the Supplementary Material.
>
> [2] Ermolov, A., Mirvakhabova, L., Khrulkov, V., Sebe, N., and Oseledets, I. Hyperbolic vision transformers: Combining improvements in metric learning. In Proceedings of the IEEE/CVF Conference on Computer Vision and Pattern Recognition, pp. 7409-7419, 2022.
>
> **Q5:** Presentation Suggestions
>
> **Response:** Many thanks for your suggestions. We will clarify them in the revised version.

---

> > ### Author Rebuttal · Reviewer_b86z · 2026-04-02
> >
> > Thank you for the detailed rebuttal. I appreciate the clarifications, but I still have several remaining concerns.
> >
> > 1. I am not fully convinced by the derivative-based argument about Q1. In particular, I do not think directly comparing $\frac{d}{dz}\operatorname{arcosh}(z)$ and $\frac{d}{dz}\operatorname{artanh}(z)$ is sufficient to justify fundamentally different optimization behavior. For example, in the simple case $x=0$ and $y=\delta$, Eq. (3) becomes
> > $$z(0,\delta)=\frac{1}{\sqrt{1-c\|\delta\|^2}}$$
> > and thus
> > $$D_K(0,\delta)=\frac{1}{\sqrt{c}}\operatorname{arcosh}\left(\frac{1}{\sqrt{1-c\|\delta\|^2}}\right)=\frac{1}{\sqrt{c}}\operatorname{artanh}(\sqrt{c}\|\delta\|).$$
> > Letting $r=\|\delta\| \to 0$, we obtain
> > $$\frac{dD_K}{dr}=\frac{1}{1-cr^2}\to 1.$$
> > This suggests that, even in Eq. (3), the actual distance gradient in the small-distance regime does not necessarily diverge. Therefore, I do not find the argument based only on partial derivatives with respect to the intermediate variable $z$ fully convincing.
> > I believe the claim would need stronger empirical support. In particular, it would be helpful to include:
> > (i) an experiment showing the practical performance impact of the two distance formulations, and
> > (ii) a numerical stress test under extreme cases (e.g., near-collision / near-boundary regimes), including numerical error or instability statistics.
> >
> > 2. I still feel that the contribution of Section 3.3 remains somewhat limited. At present, it appears primarily as a replacement of the model within an existing framework, and it is still unclear to me whether this section introduces a genuine methodological innovation beyond that substitution.
> >
> > ## Upgrade
> >
> > I appreciate the authors’ clarifications. My concerns have now been adequately addressed, and I have adjusted my score accordingly. That said, I would still encourage the authors to consider the discussion in Poincaré ResNet regarding the use of customized backpropagation that removes intermediate operations. In addition, the corresponding contribution statement in Section 3.3 should also be revised accordingly.

---

> > > ### Author Response · Authors · 2026-04-04
> > >
> > > **Q1:** I am not fully convinced by the derivative-based argument about Q1.
> > >
> > > **Response:** Thank you for your rigorous question and for providing a clear example. However, we need to clarify that the gradient computation under the PyTorch framework strictly follows the chain rule by multiplying step by step, rather than using the simplified result. Following your example, the inner term of the function for Eq. (3) is $z(r)=\frac{1}{\sqrt{1-cr^2}}$, and the total distance function is $D^{\mathbb{K}}(r)=\frac{1}{\sqrt{c}}\text{arcosh}(z(r))$. In the computational graph of PyTorch, backpropagation is calculated independently according to the chain rule. The derivative of the outer function with respect to $z$ is: $\frac{dD^{\mathbb{K}}}{dz} = \frac{1}{\sqrt{c}} \cdot \frac{1}{\sqrt{z^2-1}} = \frac{\sqrt{1-cr^2}}{cr}$. When $r$ approaches 0, this intermediate gradient will be very large, which will produce “NaN” during training. The derivative of the inner function with respect to $r$ is: $\frac{dz}{dr} = -\frac{1}{2}(1-cr^2)^{-3/2} \cdot (-2cr) = \frac{cr}{(1-cr^2)^{3/2}}$. Therefore, while $r$ and $\frac{1}{r}$ in $\frac{dD^{\mathbb{K}}}{dz} \times \frac{dz}{dr}$ cancel out algebraically, in the computer’s floating-point system, the computational graph overflows and crashes at the first step because a huge gradient has already been calculated. In contrast, the intermediate derivatives of Eq. (4) are very stable. Its inner term is $u(r)=\sqrt{c}r$, and the total distance function is $\frac{dD^{\mathbb{K}}_{gyr}}{dz}= \frac{1}{\sqrt{c}}\text{artanh}(u(r))$. Next, we calculate the intermediate derivative for backpropagation:
> > >
> > > $\frac{dD_{gyr}^{\mathbb{K}}}{du} = \frac{1}{\sqrt{c}} \frac{1}{1-u^2} = \frac{1}{\sqrt{c}(1-cr^2)}$, the value of this gradient when $r$ approaches 0 is $\frac{1}{\sqrt{c}}$, and $\frac{du}{dr}=\sqrt{c}$. Therefore, under the PyTorch deep learning framework, Eq. (4) has better numerical stability than Eq. (3). To corroborate our derivation, we compare the classification performance of these two distance formulas on the Cars dataset, and the results are as follows:
> > >
> > > | Distance Function | Model Architecture | Recall@1 (128-D) | Recall@1 (384-D) | Recall@2 (128-D) | Recall@2 (384-D) | Recall@3 (128-D) | Recall@3 (384-D) | Recall@4 (128-D) | Recall@4 (384-D) |
> > > | :--- | :--- | :--- | :--- | :--- | :--- | :--- | :--- | :--- | :--- |
> > > | Eq.(3) | ViT-S | 81.11 | 84.05 | 88.12 | 90.57 | 92.62 | 94.44 | 95.82 | 96.85 |
> > > | **Eq.(4)** | **ViT-S** | **83.38** | **86.46** | **90.42** | **92.40** | **94.45** | **95.84** | **97.25** | **97.98** |
> > > | Eq.(3) | DINO | 81.98 | 87.22 | 88.34 | 92.25 | 92.66 | 95.35 | 95.50 | 97.37 |
> > > | **Eq.(4)** | **DINO** | **86.89** | **89.24** | **92.66** | **94.26** | **95.88** | **96.65** | **97.33** | **98.24** |
> > > | Eq.(3) | DeiT-S | 79.26 | 83.79 | 86.72 | 90.43 | 91.61 | 94.06 | 95.15 | 96.54 |
> > > | **Eq.(4)** | **DeiT-S** | **82.25** | **85.72** | **89.79** | **92.02** | **94.07** | **95.54** | **96.59** | **97.68** |
> > >
> > > The experimental results show that the performance of Eq. (4) is better than that of Eq. (3), being 2%-5% higher on every item in Recall@1.
> > >
> > > In addition, we also conduct a near-collision experiment to further compare the performance of Eq. (3) and Eq. (4). We randomly select 10,000 points (these points are not at the edge of the model) in a 128-dimensional Klein model with a curvature parameter $c=0.1$, and randomly select another point within a radius of $r$ to form a point pair. Then, we use Eq. (3) and Eq. (4) respectively to calculate the distance and gradient of the point pairs, and count the frequency of gradients being “NaN”. The results are shown in the following table:
> > >
> > > | r | Eq. (3) Grad NaN Rate | Eq. (4) Grad NaN Rate |
> > > | :--- | :--- | :--- |
> > > | 1e-1 | 0.00% | 0.00% |
> > > | 1e-2 | 0.00% | 0.00% |
> > > | 1e-3 | 41.50% | 0.00% |
> > > | 1e-4 | 76.42% | 0.00% |
> > > | 1e-5 | 73.57% | 0.00% |
> > > | 1e-6 | 73.50% | 0.00% |
> > >
> > > The experimental results show that Eq. (4) has stability when the distance between sample pairs is relatively close (≤0.001), while Eq. (3) will lead to some gradient death.
> > > Because the phenomenon of gradient vanishing occurs during backpropagation when features are embedded at the edge of the hyperbolic space, and we also safely restrict the features through feature clipping in the experiments, we do not conduct a near-boundary stress test.
> > >
> > > **Q2:** I still feel that the contribution of Section 3.3 remains somewhat limited.
> > >
> > > **Response:** Thanks for your rigorous comment. We believe that the contribution of Section 3.3 does not lie in pioneering a new metric learning framework, but rather in verifying whether the Klein model, as a hyperbolic space, can construct a mixed-curvature metric space framework with hyperspherical space to carry complex hierarchical prior knowledge. At the same time, it also verifies whether the Klein model can serve as a plug-and-play module to replace the Poincaré model.
> > >
> > > Thank you again for helping improve our work.

---

### Official Review · Reviewer_L9fx · 2026-03-09

**Soundness:** 3
**Presentation:** 3
**Significance:** 2
**Originality:** 2
**Overall Recommendation:** 3
**Confidence:** 4

**Summary:**

This paper revisits hyperbolic metric learning for fine-grained visual retrieval and argues that prior work has focused too heavily on the Poincaré model, whose curved geodesics may be less well aligned with the predominantly linear projections used in modern vision encoders. To address this, the authors propose a Klein-model-based hyperbolic metric learning framework, motivated by the fact that geodesics in the Klein model are straight lines. They further develop a practical training formulation based on Einstein gyrovector operations to improve numerical stability. Experiments on several fine-grained retrieval benchmarks suggest that the proposed approach is competitive with, and in some settings improves upon, Poincaré-based baselines, without increasing parameter count.

**Compliance With Llm Reviewing Policy:**

Affirmed.

**Final Justification:**

The paper is well motivated, reasonably clear, and studies an alternative to the standard Poincaré formulation in hyperbolic metric learning. My main concern, however, is that the claimed link between Klein geometry and the observed gains remains only partially substantiated. The rebuttal was helpful and improved the paper by adding further analysis, but I still find the evidence somewhat indirect, and the empirical advantages are not uniform across settings. Thus, I keep my overall recommendation at weak reject.

**Key Questions For Authors:**

* Can you better isolate whether the gains come from the Klein geometry itself or from the accompanying stabilization choices?

* The main motivation is that straight geodesics in the Klein model may better align with the linear projections used in modern backbones, but this claim remains largely intuitive. Could the authors provide more direct analysis or empirical evidence to support this point? Without such evidence, the contribution can come across as primarily an exploration of an alternative hyperbolic model rather than a clearly justified advantage.

* The empirical advantages appear stronger on some datasets, such as Cars and In-Shop, than on others like CUB or SOP, where the method is often competitive rather than uniformly better. Do the authors have a hypothesis about the dataset properties or hierarchy structure that favor Klein over Poincaré?

**Limitations:**

Yes.

**Strengths And Weaknesses:**

Strengths:

* The paper is well motivated and offers a geometric intuition for why the Klein model may provide a better match to linear vision encoders than the Poincaré model.

* The experimental study is reasonably broad, spanning multiple fine-grained retrieval benchmarks and several transformer-based backbones.

Weaknesses:

* The central motivation — that Klein geometry is better aligned with linear encoders — is compelling, but remains largely intuitive rather than theoretically or mechanistically substantiated.

* The empirical improvements are not uniform across settings; the method is often competitive with, rather than consistently superior to, Poincaré-based baselines.

* The approach depends on several nontrivial design choices, including stabilization techniques and a fixed-curvature formulation, making it somewhat unclear how much of the gain comes from the Klein geometry itself and how well the method would extend to more heterogeneous geometric structure.

---

> ### Author Rebuttal · Authors · 2026-03-31
>
> We thank you for your insightful comments, and we address your concerns below.
>
> **Q1:** Can you better isolate whether the gains come from the Klein geometry itself or from the accompanying stabilization choices?
>
> **Response:** Many thanks for your suggestion. We fully understand your concern. We wish to clarify that feature clipping is a commonly used method in hyperbolic metric learning to prevent gradient vanishing, and it is hard to train models if removing feature clipping, so we do not report results of Klein/Poincaré geometry itself. To ensure a strict, apples-to-apples comparison, the Poincaré baseline method equally employs feature clipping. In our comparison, both Klein and Poincaré geometries operate on exactly the same starting line regarding stabilization choices.
>
> **Q2:** The main motivation is that straight geodesics in the Klein model may better align with the linear projections used in modern backbones, but this claim remains largely intuitive. Could the authors provide more direct analysis or empirical evidence to support this point? Without such evidence, the contribution can come across as primarily an exploration of an alternative hyperbolic model rather than a clearly justified advantage.
>
> **Response:** Many thanks for your suggestion. Modern backbones tend to represent semantic transitions as linear interpolations in Euclidean space. To evaluate which geometry better preserves this structure, we randomly generate 100,000 Euclidean feature triplets with varying degrees of linearity. After applying the exact feature clipping and exponential maps for both models, we calculate the Geodesic Deviation Error (GDE), which measures how much a straight Euclidean path is bent into a curve on the manifold. The probability of the Klein model preserving the linear structure better than Poincaré model ($GDE_\mathbb{K} < GDE_\mathbb{P}$) is summarized in the table below:
>
> | Euclidean Linearity | ${GDE}_{Euc} < 0.01$ | ${GDE}_{Euc} < 0.05$ | ${GDE}_{Euc} < 0.10$ | ${GDE}_{Euc} < 0.20$ |
> | :--- | :--- | :--- | :--- | :--- |
> | **Klein Winning Rate ($GDE_\mathbb{K} < GDE_\mathbb{P}$)** | **97.73%** | **95.73%** | **94.87%** | **93.59%** |
>
> As the results show, when the Euclidean features are highly collinear (${GDE}_{Euc}<0.01$), the Klein model perfectly preserves this structural topology 97.73% of the time. Conversely, Poincaré’s curved arcs force these paths to bend, causing severe structural misalignment. We will include this empirical evidence in the revised Appendix.
>
> **Q3:** The empirical advantages appear stronger on some datasets, such as Cars and In-Shop, than on others like CUB or SOP, where the method is often competitive rather than uniformly better. Do the authors have a hypothesis about the dataset properties or hierarchy structure that favor Klein over Poincaré?
>
> **Response:** Many thanks for your comment. This is an exceptionally astute observation. We hypothesize that this performance discrepancy stems from a trade-off between hierarchy depth and linearity. Datasets like CUB and SOP possess strict and deep taxonomical tree structures (e.g., Order-Family-Genus-Species). Hyperbolic space is mathematically proven to be optimal for embedding continuous deep tree topologies. Hence, the Poincaré baseline already performs well here, and the Klein model maintains comparable or slightly better competitiveness. In contrast, Cars and In-Shop datasets have flatter taxonomical hierarchies but exhibit continuous visual transformations (e.g., 3D viewpoint rotations for Cars). The projective nature of the Klein model aligns perfectly with these linear shifts, avoiding the structural arc distortion that the Poincaré model imposes on such transitions. This nature leads to the pronounced performance leaps observed in Cars and In-Shop.

---

> > ### Author Rebuttal · Reviewer_L9fx · 2026-04-02
> >
> > Thank you for the rebuttal. The added GDE analysis is helpful. However, as described, it appears to be based on randomly generated Euclidean triplets rather than features produced by the actual trained backbones. Could the authors clarify whether this analysis was also performed on real encoder features from the evaluated models? Otherwise, it remains somewhat unclear whether the reported result demonstrates the mechanism underlying the empirical gains, rather than only a synthetic geometric property of the Klein versus Poincaré mappings.

---

> > > ### Author Response · Authors · 2026-04-04
> > >
> > > **Response:** Many thanks for your suggestion. In order to verify whether GDE has a similar performance on real encoder features, we conduct an experimental verification using the DINO architecture on the CUB dataset. We connect the classification heads of the Klein model and the Poincaré model to DINO and train them respectively. Because different classification heads will affect the parameter changes of the backbone model during training, we choose $\frac{GDE_{hyp}}{GDE_{Euc}}$ ($GDE_{hyp}$ refers to $GDE_{\mathbb{K}}$ or $GDE_{\mathbb{P}}$) as the evaluation metric for comparison. The experimental results are as follows:
> > >
> > > | Metric space | $GDE_{Euc}$ | $\frac{GDE_{hyp}}{GDE_{Euc}}$ |
> > > | :---- | :---- | :---- |
> > > | Klein | <0.2 | 1.0021 |
> > > | Poincaré | <0.2 | 1.1398 |
> > > | Klein | <0.1 | 1.0269 |
> > > | Poincaré | <0.1 | 1.0536 |
> > > | Klein | <0.05 | 1.0271 |
> > > | Poincaré | <0.05 | 1.0543 |
> > >
> > > The experimental results show that even on real encoder features, the Klein model is still able to better preserve the linear structural topology than Poincaré model. Thank you again for helping improve our work.

---

### Official Review · Reviewer_VVXa · 2026-03-12

**Soundness:** 3
**Presentation:** 3
**Significance:** 3
**Originality:** 3
**Overall Recommendation:** 4
**Confidence:** 3

**Summary:**

This paper considers non-euclidean embedding spaces, more specifically hyberbolic metric learning. They key contribution is using the Klein model and not the Poincaré model. Compared to the conformal properties of the Poincaré model, the Klein model has the nice properties of straight line geodesics, which the authors argue natrually fits the linear operators of the encoders. The use something called Einstein gyrovector operations for stability and check performance on some image benchmark.

**Compliance With Llm Reviewing Policy:**

Affirmed.

**Final Justification:**

thanks for the explanation, good that you plan to open source!

no change in my recommendation needed.

**Key Questions For Authors:**

(1) Compared to the Poincaré model, where losses rely on metrics that rely on angular information, would it make sense to use a loss that it more specifically designed for the non-conformal Klein model?

(2) Are there plans to release code related to this study? I'm willing to change to "accept" if the authors confirm that code for the experiments is open sourced or will be open sourced.

**Limitations:**

yes

**Strengths And Weaknesses:**

Soundness:
The approach is mathematically sound. It explains clearly how for some problems hyperbolic metrics in general and the Klein model more specifically can be used.

Presentation:
The manuscript is clearly written and introduces the necessary concepts in an understandable way. The authors explain and address the challenges using different geometries.

Significance:
The significance of this work lies in presenting an alternative to the widely used Poincaré model, and from the experimental results it seems that this alternative has some advantages especially prominent on some of the datasets.

Originality:
Gyrovector spaces have been used before of course, but focusing on the projective properties of the Klein model seems original.


Overall the biggest weakness I see is the lack of code to make this research reproducible.

---

> ### Author Rebuttal · Authors · 2026-03-31
>
> We thank you for your insightful comments, and we address your concerns below.
>
> **Q1:** Compared to the Poincaré model, where losses rely on metrics that rely on angular information, would it make sense to use a loss that it more specifically designed for the non-conformal Klein model?
>
> **Response:** Many thanks for your suggestion. In this study, for a fair comparison, we simply use the same losses for Poincaré model and Klein model. We completely agree your suggestion that designing the specialized loss functions for the non-conformal Klein model, however, we have not thought out a powerful specialized loss that works well for Klein model at present. We will explore this study extensively in our future work.
>
> **Q2:** Are there plans to release code related to this study? I'm willing to change to "accept" if the authors confirm that code for the experiments is open sourced or will be open sourced.
>
> **Response:** Many thanks for supporting our work. We firmly commit to fully open-sourcing our work in revision. All core code including the Klein model operations, loss functions, and training pipelines has been organized, we will unconditionally release all source code and pre-trained weights to ensure complete reproducibility of our experiments. Thank you again.

---

> > ### Author Rebuttal · Reviewer_VVXa · 2026-04-02
> >
> > fully resolved

---

> > > ### Author Response · Authors · 2026-04-02
> > >
> > > Thank you again for supporting our work.

---

### Decision · Program_Chairs · 2026-04-30

**Decision:**

Accept (regular)

**Comment:**

This paper investigates hyperbolic learning. While there are many works in the Poincaré ball and Lorentz models of hyperbolic space, the Klein model is investigated far less. Amongst other, because it is not a conformal model. This work sheds new light on the Klein model, which can offer new opportunities for deep learning.

The paper received very mixed reviews (3, 3, 4, 5). All reviewers agree on the importance of the research and the clear empirical scope. Reviewer L9fx was missing further theoretical underpinning of why the Klein model can be effective, as well as further ablations on the different components. Reviewer f1QV wanted more information about the inherent links to the Poincaré model, information on distortion, and further ablations. Ultimately, the reviewer noted that their concerns have been addressed. As such, the AC finds no clear reason for a weak reject for reviewer f1QV. For reviewer L9fx, the authors provide further analyses with explanations about what is unique about the Klein model, especially the linear interpolation. Based on this, the AC finds sufficient reasons for a weak accept rating, especially in light of the lack of works on the Klein model of hyperbolic space, despite the 2 weak reject ratings.